# Autobahn: Automorphism-based Graph Neural Nets

**Erik H. Thiede,[1*] Wenda Zhou,[12†] Risi Kondor[134‡]**

[1] Center for Computational Mathematics, Flatiron Institute, New York NY 10010
[2] Center for Data Science, New York University, New York NY 10011
[3] Department of Computer Science, University of Chicago, Chicago IL 60637
[4] Department of Statistics, University of Chicago, Chicago IL 60637
[*]ehthiede@flatironinstitute.org, [†]wz2247@nyu.edu, [‡]risi@uchicago.edu

## Abstract

We introduce Automorphism-based graph neural networks (Autobahn), a new family of graph neural networks. In an Autobahn, we decompose the graph into a collection of subgraphs and apply local convolutions that are equivariant to each subgraph's automorphism group. Specific choices of local neighborhoods and subgraphs recover existing architectures such as message passing neural networks. Our formalism also encompasses novel architectures: as an example, we introduce a graph neural network that decomposes the graph into paths and cycles. The resulting convolutions reflect the natural way that parts of the graph can transform, preserving the intuitive meaning of convolution without sacrificing global permutation equivariance. We validate our approach by applying Autobahn to molecular graphs, where it achieves results competitive with state-of-the-art message passing algorithms.

## 1 Introduction

The successes of artificial neural networks in domains such as computer vision and natural language processing have inspired substantial interest in developing neural architectures on graphs. Since graphs naturally capture relational information, graph-structured data appears in a myriad of fields. However, working with graph data raises new problems. Chief among them is the problem of graph isomorphism: for the output of our neural network to be reliable, it is critical that the network gives the same result independent of trivial changes in graph representation such as permutation of nodes.

Considerable effort has gone into constructing neural network architectures that obey this constraint[1–5]. Arguably, the most popular approach is to construct *Message-Passing Neural Networks (MPNNs)*[6, 7]. In each layer of an MPNN, every node aggregates the activations of its neighbors in a permutation invariant manner and applies a linear mixing and nonlinearity to the resulting vector. While subsequent architectures have built on this paradigm, e.g., by improving activations on nodes and graph edges [8, 9], the core paradigm of repeatedly pooling information from neighboring nodes has remained. These architectures are memory efficient, intuitively appealing, and respect the graph's symmetry under permutation of its nodes. However, practical results have shown that they can oversmooth signals [10] and theoretical work has shown that they have trouble distinguishing certain graphs and counting substructures [8, 11–13]. Moreover, MPNNs do not use the graph's topology to its fullest extent. For instance, applying an MPNN to highly structured graphs such as grid graphs does not recover powerful known architectures such as convolutional neural networks. It is also not clear how to best adapt MPNNs to families of graphs with radically different topologies: MPNNs for citation graphs and molecular graphs are constructed in largely the same way. This suggests that it should be possible to construct more expressive graph neural networks by directly leveraging the graph's structure.

In this work, we introduce a new framework for constructing graph neural networks, *Automorphism-based Neural Networks* (Autobahn). Our research is motivated by our goal of designing neural

35th Conference on Neural Information Processing Systems (NeurIPS 2021).

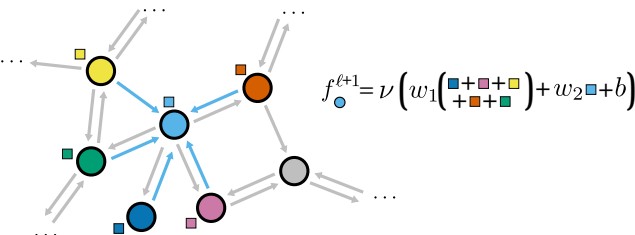

$$f_\circ^{\ell+1} = \nu \left( w_1 \left( \begin{smallmatrix} \blacksquare + \square + \square \\ + \blacksquare + \blacksquare \end{smallmatrix} \right) + w_2 \blacksquare + b \right)$$

Figure 1: Visualization of a single neuron of a simple message passing neural network. Each node aggregates the features from neighboring nodes using a permutation-invariant operation (we use summation for simplicity), applies learned weight matrices and biases and finally a nonlinearity.

networks that can learn the properties of small organic molecules accurately enough to make a significant contribution to drug discovery and materials design [14–17]. The properties of these molecules depend crucially on multi-atom substructures, making the difficulties MPNNs have in recognizing substructures a critical problem. We realized that we could circumvent this limitation by making graph substructures *themselves* the fundamental units of computation. The symmetries of our substructures then inform the computation. For instance, the benzene molecule forms a ring of six atoms and is a common subunit in larger molecules. On this ring we have a very natural and mathematically rigorous notion of convolution: convolution on a one-dimensional, periodic domain. This symmetry is encoded by a graph's automorphism group: the group that reflects our substructures' internal symmetries. Our networks directly leverage the automorphism group of subgraphs to construct flexible, efficient neurons. Message passing neural networks arise naturally in this framework when the substructures used are local star graphs, and applying Autobahn to grid graphs can recover standard convolutional architectures such as steerable CNNs. More generally, Autobahn gives practitioners the tools to build bespoke graph neural networks whose substructures reflect their domain knowledge. As an example, in Section 6, we present a novel architecture outside of the message-passing paradigm that nevertheless achieves performance competitive with state-of-the-art MPNNs on molecular learning tasks.

## 2 Graph Neural Networks

Neural Networks operate by composing several learned mappings known as "layers". Denoting the $\ell$'th layer in the network as $\phi_\ell$, the functional form of a neural network $\Phi$ can be written as

$$\Phi = \phi_L \circ \phi_{L-1} \circ \ldots \circ \phi_1.$$

Each layer is typically constructed from a collection of parts, the titular "neurons". We denote the $i$'th neuron in the $\ell$'th layer as $\mathfrak{n}_i^\ell$, and denote its output (the "activation" of the neuron) as $f_i^\ell$. Architectures differ primarily in how the neurons are constructed and how their inputs and outputs are combined.

When constructing a neural network that operates on graph data, care must be taken to preserve the input graphs' natural symmetries under permutation. Let $\mathcal{G}$ be a graph with node set $\{v_1, \ldots, v_n\}$, adjacency matrix $A \in \mathbb{R}^{n \times n}$ and $d$-dimensional node labels $b_i \in \mathbb{R}^d$ stacked into a matrix $B \in \mathbb{R}^{n \times d}$. Permuting the numbering of the nodes of $\mathcal{G}$ by some permutation $\sigma \colon \{1, 2, \ldots, n\} \to \{1, 2, \ldots, n\}$ transforms

$$A \mapsto A^\sigma \qquad\qquad A_{i,j}^\sigma = A_{\sigma^{-1}(i), \sigma^{-1}(j)}, \qquad\qquad (1)$$

and

$$B \mapsto B^\sigma \qquad\qquad B_{i,n}^\sigma = B_{\sigma^{-1}(i), n}. \qquad\qquad (2)$$

This transformation does not change the actual topology of $\mathcal{G}$. Consequently, a fundamental requirement on graph neural networks is that they be invariant with respect to such permutations.

### 2.1 Message-Passing Neural Networks

Message-passing neural networks (MPNNs) have emerged as the dominant paradigm for constructing neural networks on graphs. Every neuron in an MPNN corresponds to a single node in the graph. Neurons aggregate features from neighboring nodes by passing them through a permutation invariant function. They then combine the result with the node's original message, and pass the result through a learned linear function and nonlinearity (Figure 1)[8]. Since information is transmitted over the graph's topology, MPNNs are not confounded by permutations of the graph. Specific architectures

may differ in the details of the aggregation function [6, 8, 18, 19], may include additional terms to account for edge features [8, 9], may use complex transformations to construct the node features encoding local structure [20, 21], or may augment the graph with additional nodes [22, 23].

# 3 Permutation Equivariance

As discussed in the Introduction, MPNNs have fundamental limits to their expressiveness. To construct more powerful neural networks, we turn to the general formalism of group equivariant networks [24–27]. Our desire that permutations of the input graph leave our network's output unaffected is formalized by the notion of group-invariance. Let us assume that our input data lives in a space $X$ that is acted on by a group $G$, and for all $g \in G$ denote the associated group action on $X$ by $T_g$. The invariance constraint amounts to requiring:

$$\Phi = \Phi \circ T_g \qquad\qquad \forall g \in G. \tag{3}$$

One way to satisfy this would be to require that each layer $\phi_\ell$ be fully invariant to $G$. However, in practice this condition can be extremely restrictive. For this reason, networks commonly use group *equivariant* layers. Let $X$ and $Y$ be the input and output spaces of $\phi_\ell$, with group actions $T_g$ and $T'_g$, respectively. We say $\phi_\ell$ is equivariant to $G$ if it obeys

$$T'_g \circ \phi_\ell = \phi_\ell \circ T_g \qquad\qquad \forall g \in G. \tag{4}$$

This condition is weaker than invariance: we recover invariance when $T'_g$ maps every element of $Y$ to itself. Moreover, it is simple to show that the composition of two equivariant layers is also equivariant. Consequently, in all layers but the last we can enforce the weaker condition of equivariance and merely enforce invariance in the final layer. In the case of graph neural networks, the relevant group is the group of permutations: $\mathbb{S}_n$ (called the symmetric group of degree $n$).

## 3.1 Permutation-equivariant networks

Recent work has developed a generic recipe for constructing group equivariant networks. In this formalism, any object that transforms under a group action is treated as a function on the group [24] (see Section 1 in the supplement for a brief review). This allows all layers equivariant to the group to be treated using the same formalism, independently of how inputs and outputs transform. In particular, it can be shown that the only group-equivariant linear operation possible is a generalized notion of group convolution. For discrete groups, this convolution can be written as:

$$(f * w)(u) = \sum_{v \in G} f(uv^{-1}) \, w(v). \tag{5}$$

To construct an equivariant neuron, we apply (5) to convolve our input activation $f^{\ell-1}$ with a learned weight function $w$, add a bias, and then apply a fixed equivariant nonlinearity, $\nu$. Applying this approach to specific groups recovers the standard convolutional layers used in convolutional neural networks (CNNs). For instance, applying (5) to the cyclic group of order $n$ gives

$$(f * w)_j = \sum_{k=0}^{n} f(r^{j-k}) \, w(r^k), \tag{6}$$

where $r$ is the group element corresponding to rotation by $360/n$ degrees. Similarly, applying (5) to one-dimensional or two-dimensional discrete translation groups recovers the standard convolutions used for image processing.

Instantiating this theory with the symmetric group has been successfully used to construct permutation equivariant networks for learning tasks defined on sets [28] and for graph neural networks [29–32]. However, enforcing equivariance to all permutations can be very restrictive. As an example, consider a layer whose domain and range transform according to (2) with $c_{in}$ incoming and $c_{out}$ outgoing channels. Let $w_1, w_2 \in \mathbb{R}^{c_{out} \times c_{in}}$ be learned weight matrices, and $f_j \in \mathbb{R}^{c_{in}}$ be the vector of incoming activations corresponding to node $j$. The most general possible convolution is then [28]

$$(f * w)_i = w_1 f_i + w_2 \sum_{j=1}^{n} f_j, \tag{7}$$

but this is quite a weak model as activations from different nodes only interact through their sum. Consequently, relationships between nodes can only be captured in aggregate.

---

**Algorithm 1** Automorphism-based Neuron

---

**Input:**
    $f^{\ell-1}$                ▷ Incoming activation associated with $\mathcal{G}$
    $A_{\mathcal{G}}$                           ▷ Adjacency matrix of $\mathcal{G}$
    $A_{\mathcal{T}}$              ▷ Adjacency matrix of the saved template graph.
1: Find $\mu \in \mathbb{S}_n$ such that $A_{\mathcal{G}}^{\mu} = A_{\mathcal{T}}$.
2: $(f^{\ell-1})^{\mu} \leftarrow T_{\mu}(f^{\ell-1})$                ▷ Apply $\mu$ to incoming activation.
3: $(f^{\ell})^{\mu} \leftarrow \nu\left((f^{\ell-1})^{\mu} * w + b\right)$         ▷ Convolution is over $\mathrm{Aut}(\mathcal{T})$.
4: $f^{\ell} \leftarrow T'_{\mu^{-1}}((f^{\ell})^{\mu})$             ▷ Map output to original ordering
**Return:** $f^{\ell}$

---

To address this fundamental limitation, several recent works considered improving the expressivity of MPNNs by defining higher order activations corresponding to pairs, triplets, or, in general, $k$–tuples of nodes [29, 32, 33]. Mathematically, this requires considering not just (1) and (2), but the action of the symmetric group on $k$'th order tensors, $A_{i_1,\ldots,i_k}^{\sigma} = A_{\sigma^{-1}(i_1),\ldots,\sigma^{-1}(i_k)}$. However, this can be prohibitively expensive for many nodes. For instance, organic chemistry depends crucially on the existence of aromatic rings, typically of six atoms. Manipulating sixth order tensors would be extremely costly.

## 4   Permutation-Equivariant Neurons using Automorphism

The key theoretical idea underlying our work is that to construct more flexible neural networks, we can exploit the graph topology itself. In particular, the local adjacency matrix itself can be used to judiciously break permutation symmetry, allowing us to identify nodes up to the symmetries of $A$. Letting $\mathcal{G}$ be a graph as before, the automorphism group $\mathrm{Aut}(\mathcal{G})$ is:

$$\mathrm{Aut}(\mathcal{G}) = \left\{\, \sigma \in \mathbb{S}_n \,|\, A^{\sigma} = A \,\right\}, \tag{8}$$

where $A^{\sigma}$ is defined as in (1). Figure 2a shows the automorphism group of four example graphs.

If $\mathcal{G}$ and $\mathcal{G}'$ are two isomorphic graphs, then each node or edge in $\mathcal{G}'$ can always be matched to a node or edge in $\mathcal{G}$ *up to a permutation in* $\mathrm{Aut}(\mathcal{G})$. If every graph our network observed was in the same isomorphism class, we could construct a neuron in a permutation-equivariant neural network by matching $\mathcal{G}$ to a template graph $\mathcal{T}$ and convolving over $\mathrm{Aut}(\mathcal{T})$. We give pseudocode for such a neuron, which we call an "Automorphism-based neuron," in Algorithm 1. Note that although the convolution itself is only equivariant to $\mathrm{Aut}(\mathcal{T})$, the entire neuron is permutation-equivariant. A formal proof of equivariance is given in Section 2 of the Supplement.

For a given input and output space, neurons constructed using Algorithm 1 are more flexible than a neuron constructed using only $\mathbb{S}_n$ convolution. As an example, consider a neuron operating on graphs isomorphic to the top left graph in Figure 2a and whose input and output features are a single channel of node features. An $\mathbb{S}_n$-convolution would operate according to (7): Each output node feature would see only the corresponding input feature and an aggregate of all other node features. In contrast, Algorithm 1 completely canonicalizes the graph since this particular graph has no non-trivial automorphisms. Consequently, we do not need to worry about enforcing symmetry and we can simply run a fully connected layer: a richer representation, since it does not require the same degree of "parameter sharing" as (7). Similarly, Figure 2b depicts an analogous neuron constructed in the presence of cyclic graph symmetry. In this case, the matching can be performed up to a cyclic permutation, so convolution must be equivariant to the graph's automorphism group, $C_6$.

This general strategy was first described by de Haan et al. [34]. While the increased flexibility was noted, the authors also observed rightfully that this strategy is not directly practical. Most graph learning problems consist either of many graphs from different isomorphism classes or a single large graph that is only partially known. In the former case, we would have to construct one network for every isomorphism class and each network would only see a fraction of the data. In the latter, the isomorphism class of the network would be unknown.

## 5   Autobahn

To motivate our approach we consider MPNNs from the perspective of Algorithm 1. Looking at Figure 1 we see that the message-passing procedure itself forms a star graph, where the leaves of

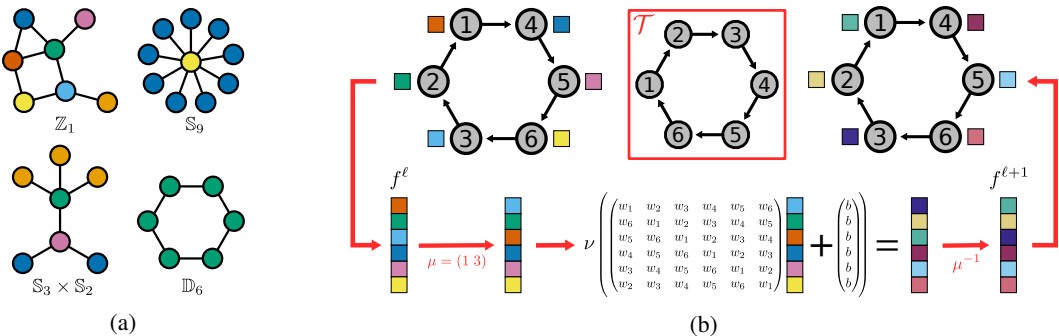

(a)                  (b)

Figure 2: Figures visualizing the automorphism group of a graph and its use in graph learning. (a) Four graphs and their automorphism groups. In each graph, nodes in the same orbit of the graph's automorphism group are the same color. (b) A neuron constructed by applying Algorithm 1 to a cyclic directed graph. We consider the simplified setting where the layer operates only on a single channel of node features. Note that the matching step can only be accomplished up to an element in the graph's automorphism group: the cyclic group of order six, $\mathbb{C}_6$.

the star correspond to the neighboring nodes. The automorphism group of a star graph is the set of all permutations that swap the star's leaves. This is precisely the group structure of a single MPNN neuron. Since we apply a permutation-invariant aggregation function, if we were to permute a neuron's input data between the leaves of the star, the MPNN neuron would be unaffected. However, permuting the input features for the central node with one of its neighbors would "break" the MPNN.

This suggests a natural generalization of MPNNs. In every layer, we decompose a graph into a collection of subgraphs known as *local graphs* that are isomorphic to a pre-selected template graph. Although in worst case this may be polynomially expensive, for "real world" graphs we expect that it can be solved efficiently. In particular, one can leverage the well-developed literature of efficient search heuristics [35–38]. We next construct a permutation-equivariant neuron on each local graph, denoting the local graph of neuron $\mathfrak{n}_j^\ell$ as $\mathcal{G}_j^\ell$. Each neuron operates by aggregating information from overlapping local graphs and then applying Algorithm 1 to the result. Because of the important role played by convolutions over subgraphs' automorphism groups, we refer to the resulting networks as Automorphism-based Neural Networks, or *Autobahns*.

MPNNs are not the only commonly used algorithm that can be recovered by the Autobahn formalism. In Section 3 in the supplement we show that using a grid graph template on a larger grid graph recovers steerable CNNs [26][1]. The fact that Autobahn networks naturally recover these architectures when MPNNs do not suggests that considering the local automorphism group is a productive direction for incorporating graph structure.

Our work builds on a considerable body of literature on constructing neural networks with higher-order activations [33, 34, 39, 40]. Arguably, our work is most closely related to natural graph networks (NGNs), the formalism proposed in [34]. However, NGNs associate neurons with neighborhoods of individual nodes and edges, making the direct use of automorphic convolutions impractical for generic graphs. The specific architecture proposed in [34] instead combined multiple message passing networks, each applied to a local neighborhood of the graph. In contrast, Autobahn associates neurons with subgraphs instead of specific nodes or edges. To our knowledge, this is the first work to explicitly consider constructing neurons equivariant to the automorphism group of subgraphs. This additional flexibility allows us to build practical networks using the convolutions described in Section 4

In the discussion that follows, we give a generic treatment of each step in the network, followed by a full description of an Autobahn layer. We do not specify a specific form for the activations: they may correspond to individual nodes, edges, hyper-edges, or be delocalized over the entire local graph. This is in keeping with our philosophy of giving a flexible recipe that practitioners can tailor to their specific problems using their domain knowledge.

---

[1]Standard CNNs further break the symmetry of the grid graph by introducing a notion of up/down and left/right. However, if we introduce a notion of edge "color" to distinguish horizontal from vertical edges and extend our definition of automorphism to include color, Autobahn can recover CNNs as well: see Section 3 in the Supplement.

## 5.1 Convolutions using the Automorphism Group

The convolutions in Autobahn proceed by applying Algorithm 1 to each neuron's *local* graph. The precise form of the convolution will depend on how the activation transforms under permutation. In [24] it was observed that for any compact group, one could construct the appropriate notion of group convolution by expressing the activation in the group's Fourier space and applying the noncommmutative generalization of the convolution theorem. Subsequent work has lead to software libraries that convolve over arbitrary finite groups [41]. Moreover, for specific groups simple convolutions are either known or intuitive to derive. For instance, in Section 6, our architecture uses directed cycle and path graphs as templates. There, group convolutions can be performed using well-known one-dimensional convolutions such as (6).

## 5.2 Transferring information across neurons

To build a rich representation of the structure graph we must be able to transmit information between neurons operating on different local graphs. In MPNNs and CNNs, each neuron pools information into the central node before transmitting to its neighbors. This simplifies the task of transmitting information to other neurons, as the output of neuron $\mathfrak{n}_j^\ell$ becomes a node feature for neuron $\mathfrak{n}_k^{\ell+1}$. However, this strategy does not necessarily work for the neurons in Autobahn: our local graphs may not have a central node. Even if they do, collapsing each neuron's output into a single node could limit our network's expressivity, as it prevents neurons from transmitting equivariant information such as the hidden representations of multiple nodes or (hyper)edges.

Instead, we observe that any part of an activation that corresponds to nodes shared between two local graphs can be freely copied between the associated neurons. To transmit information from $\mathfrak{n}_j^{\ell-1}$ to $\mathfrak{n}_j^\ell$, we define two operations, narrowing and promotion, that extend this copying procedure to arbitrary activations. Narrowing compresses the output of $\mathfrak{n}_j^{\ell-1}$ into the intersection of the two local graphs, and promotion expands the result to the local graph of $\mathfrak{n}_j^\ell$.

To ensure that narrowing and promotion are correctly defined for all activations regardless of their specific group actions, we employ the formalism from [24], where activations are treated as functions on the symmetric group. Specifically, the input activation and narrowed activation on $m$ and $k$ nodes respectively are identified with functions from $\mathbb{S}_m \to \mathbb{R}$ and $\mathbb{S}_k \to \mathbb{R}$. We discuss special cases after the definitions, and depict a specific example operating on edge features in Figure 3.

### 5.2.1 Narrowing

Narrowing takes an activation $f$ that transforms under permutation of a given set of $m$ nodes and converts it into a function that transforms only with respect to a subset of $k < m$ nodes, $\{v_{i_1}, \ldots, v_{i_k}\}$. To construct our narrowed function, we apply an arbitrary permutation that "picks out" the nodes indexed by $\{i_1, \ldots, i_k\}$ by sending them to the first $k$ positions. (Note this implicitly orders these nodes.) Subsequent permutations of $\{1, 2, \ldots, k\}$ permute our specially chosen nodes amongst each other and permutations of $\{k+1, \ldots, m\}$ permute the other, less desirable nodes. Narrowing exploits this to construct a function on $\mathbb{S}_k$: we apply the corresponding group element in $\mathbb{S}_k$ to the first $k$ positions, average over all permutations of the last $m - k$, and read off the result.

**Definition 1.** *Let $(i_1, \ldots, i_k)$ be an ordered subset of $\{1, 2, \ldots, m\}$ and $t$ be an (arbitrarily chosen) permutation such that*

$$t(i_p) = p \qquad\qquad \forall p \in 1, \ldots, k. \tag{9}$$

*For all $u \in \mathbb{S}_k$ let $\acute{u} \in \mathbb{S}_m$ be the permutation that applies $u$ to the first $k$ elements and for all $s \in \mathbb{S}_{m-k}$ let $\grave{s} \in \mathbb{S}_m$ be the permutation that applies $s$ to the last $m - k$ elements. Given $f \colon \mathbb{S}_m \to \mathbb{R}^d$ we define the **narrowing** of $f$ to $(i_1, \ldots, i_k)$ as the function:*

$$f{\downarrow}_{(i_1 \ldots i_k)}(u) = (m-k)!^{-1} \sum_{s \in \mathbb{S}_{m-k}} f(\acute{u}\grave{s}t). \tag{10}$$

Narrowing obeys a notion of equivariance for permutations restricted to the local graph. Let $\sigma$ be a permutation that sends $\{i_1, \ldots, i_k\}$ to itself. Narrowing then obeys:

$$(f{\downarrow}_{(i_1 \ldots i_k)})^{\sigma'} = (f^\sigma){\downarrow}_{(i_1 \ldots i_k)},$$

where $\sigma'$ is the permutation in $\mathbb{S}_k$ obeying:

$$\sigma'(p) = q \qquad \Longleftrightarrow \qquad \sigma(i_p) = i_q. \tag{11}$$

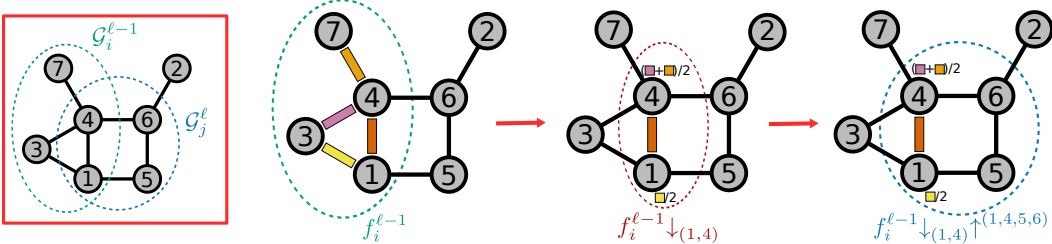

Figure 3: Example demonstrating how narrowing and promotion transfer information between local graphs. For concreteness, we assume the incoming activation $f_i^{\ell-1}$ is a collection of edge features. We first narrow to the nodes shared between $\mathcal{G}_i^{\ell-1}$ and $\mathcal{G}_j^{\ell}$. For the edge features used here, this corresponds to averaging over nodes 3 and 7; the edge inside the restriction is simply copied. The results are then placed into the appropriate position in the local graph of the output.

and the superscripts denote the transformation of the function under permutation,

$$f^{\sigma}(g) = f(\sigma g) \qquad \forall g \in \mathbb{S}_m \ (\mathbb{S}_k).$$

When applied to a collection of node features, narrowing simply saves the features in nodes in $i_1 \ldots i_k$ and the average feature and then discards the rest. More generally, if the activation is a multi-index tensor whose indices correspond to individual nodes, narrowing forms new tensors by averaging over the nodes *not* in $\{i_1 \ldots i_k\}$.

### 5.2.2 Promotion

Promotion is the opposite of narrowing in that it takes a function $g \colon \mathbb{S}_k \to \mathbb{R}^d$ and extends it to a function on $\mathbb{S}_m$. We therefore apply the same construction as in Definition 1 in reverse.

**Definition 2.** *Let $(j_1, \ldots, j_m)$ be an ordered set of indices with an ordered subset $(i_1, \ldots, i_k)$. Let $u$, $s$, $t$, $\acute{u}$, and $\grave{s}$ be as in Definition 1. Given a function $g \colon \mathbb{S}_k \to \mathbb{R}^d$, we define the **promotion** of $g$ to $\mathbb{S}_m$ as the function:*

$$g\!\uparrow^{(j_1 \ldots j_m)}(\tau) = \begin{cases} g(u) & \text{if there exist } u \in \mathbb{S}_k, \ s \in \mathbb{S}_{m-k} \text{ such that } \tau = \acute{u}\grave{s}t, \\ 0 & \text{otherwise.} \end{cases} \tag{12}$$

In Section 4 of the supplement we show that any such $u$ and $s$ are unique and consequently our definition is independent of the choice of $u$ and $s$. Narrowing is the pseudoinverse of promotion in the sense that for any $g \colon \mathbb{S}_k \to \mathbb{R}^d$

$$g\!\uparrow^{(j_1 \ldots j_m)}\!\downarrow_{(i_1 \ldots i_k)} = g.$$

In contrast, narrowing followed by promotion is typically a lossy operation that does not preserve a function. Similarly to narrowing, promotion obeys the equivariance property:

$$(g^{\sigma'})\!\uparrow^{(j_1 \ldots j_m)} = (g\!\uparrow^{(j_1 \ldots j_m)})^{\sigma},$$

where $\sigma$ and $\sigma'$ are defined as in (11). In the case of node features, promotion simply copies the node features into the new local graph. For a multi-index tensors whose indices correspond to individual nodes, promotion zero-pads the tensor, adding indices for the new nodes.

### 5.3 Autobahn neurons

Stated most generally, an Autobahn neuron $\mathfrak{n}_j^{\ell}$ operates as follows. Let $\mathfrak{n}_j^{\ell}$ be a neuron whose local graph $\mathcal{G}_j^{\ell}$ is defined on the nodes $\{v_{a_1}, \ldots, v_{a_m}\}$. Denote by $f_{s_1}^{\ell-1}, \ldots, f_{s_p}^{\ell-1}$ the activations of the neurons in the previous layers whose local graphs overlap with $\mathcal{G}_j^{\ell}$. We denote the nodes in the local graph of the $z$'th overlapping neuron by $(v_{a_1^z}, \ldots, v_{a_{m_z}^z})$ and define the intersections:

$$\{b_1^z, \ldots, b_{k_z}^z\} = \{a_1, \ldots, a_m\} \cap \{a_1^z, \ldots, a_{m_z}^z\}.$$

The operation performed by $\mathfrak{n}_j^{\ell}$ in an Autobahn can then be summarized as follows:

T1. Narrow each incoming activation $f_{s_z}^{\ell-1}$ to the corresponding intersection to get $f_{s_z}^{\ell-1}\!\downarrow_{(b_1^z \ldots b_{k_z}^z)}$.

T2. Promote each of these to $(a_1, \ldots, a_m)$:

$$\tilde{f}_{s_z} = f_{s_z}^{\ell-1}\!\downarrow_{(b_1^z \ldots b_{k_z}^z)}\!\uparrow^{(a_1 \ldots a_m)}.$$

Note that each $\tilde{f}_{s_z}$ is $(a_1, \ldots, a_m)$-permutation equivariant.

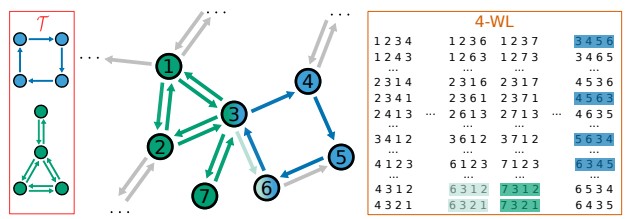

Figure 4: Visual comparison of the activations in Autobahn with the data structures used in the $k$-WL algorithm and in a $k$'th order graph neural network. Whereas the $k$-WL and $k$'th order GNN operate over all possible ordered sequences of $k$ nodes (here $k = 4$), Autobahn specifically targets sequences believed to be important by identifying them through the isomorphism class of the corresponding subgraph. Here, one subgraph is isomorphic to the blue template and two are isomorphic to the green. Sequences corresponding to their automorphism groups are highlighted in the corresponding color. Using subsets of sequences corresponding to specific substructures allows Autobahn to perform higher-order computation without the combinatorial explosion in cost.

T3. Combine the results into a single function $\tilde{f}$ by applying an aggregation function that is invariant to permutations of the set $\{\tilde{f}_1, \ldots, \tilde{f}_p\}$ within itself (for instance, averaging).

T4. Apply one or more convolutions and nonlinearities over the local graph's automorphism group as described in Algorithm 1.

Sufficient conditions for the resulting network to obey global permutation equivariance are given below.

**Theorem 1.** *Let $\mathfrak{n}_j^\ell$ be an Autobahn neuron in a neural network operating on a graph $\mathcal{G}$. Let $\mathcal{G}_j^\ell$ be the local graph of $\mathfrak{n}_j^\ell$ and denote $\mathcal{G}_j^\ell$'s node set as $\{v_{a_1}, \ldots, v_{a_m}\} \subset \{v_1, \ldots, v_n\}$ and its edges as $\mathcal{E}_j^\ell = \{e_{kl}\}_{k,l \in \{v_{a_1}, \ldots, v_{a_m}\}}$. If the following three conditions hold then the resulting Autobahn obeys permutation equivariance.*

1. *For any permutation $\sigma \in \mathbb{S}_n$ applied to $\mathcal{G}$, the resulting new network $\Phi'$ will have a neuron $\mathfrak{n}'^\ell_{j'}$ with the same parameters that operates on a graph $\mathcal{G}_j^{\ell'}$. The nodes of $\mathcal{G}_j^{\ell'}$ are $\{v_{\sigma(a_1)}, \ldots, v_{\sigma(a_m)}\}$ and its edges are $\{e_{\sigma(k)\sigma(l)} \mid e_{kl} \in \mathcal{E}_j^\ell\}$.*
2. *The output of the neuron is invariant with respect to all that permutations of $\mathcal{G}$ that leave the nodes $\{v_{a_1}, \ldots, v_{a_m}\}$ in place.*
3. *The output of the neuron is equivariant to all permutations of the set $\{v_{a_1}, \ldots, v_{a_m}\}$ within itself.*

A proof is given in Section 5 of the supplement.

### 5.4 Expressivity of Autobahn

To further understand the capabilities of Autobahn we will compare its theoretical expressivity to that of other graph neural networks. First, we analyze Autobahn in the context of the $k$-Weisfeiler-Lehman ($k$-WL) algorithm and the $k$'th order network proposed in [30]. Next, we compare Autobahn to the Graph Substructure Networks (GSN) proposed in [21].

A common tool used to analyze the expressivity of message-passing neural networks is comparison against the $k$-WL algorithm [42, 43]. Here, information is repeatedly transferred between all possible ordered sets of $k$ nodes and if the output differs between two graphs then they are not isomorphic. It has since been determined that most message-passing neural networks are limited in their expressive power by the 2-WL algorithm, meaning certain graphs are fundamentally indistinguishable by MPNNs [8, 39]. The WL algorithm is also closely related to the $k$'th order graph neural network from [30], where activations correspond to all possible ordered sets of $k$ nodes and information is transferred between sets using tensor expansions and contractions. This process corresponds to convolutions over all permutations of the graph's nodes [29]. Here, a $k$'th order network has the expressive power of the $k$-WL. However, the size of the activations grows combinatorially as we increase $k$, making the network infeasible for all but small values of $k$.

Autobahn, in contrast, constructs activations on specific subgraphs, corresponding to specifically chosen ordered sets of nodes. As depicted in Figure 4, this makes the Autobahn activation sparse in the set of all possible node sets. If these sets are chosen well, the Autobahn network can hopefully leverage the power of higher-order computation without incurring a combinatorial increase in cost. In Subsection 6.1 of the Supplement we formalize this connection by showing that the operations in

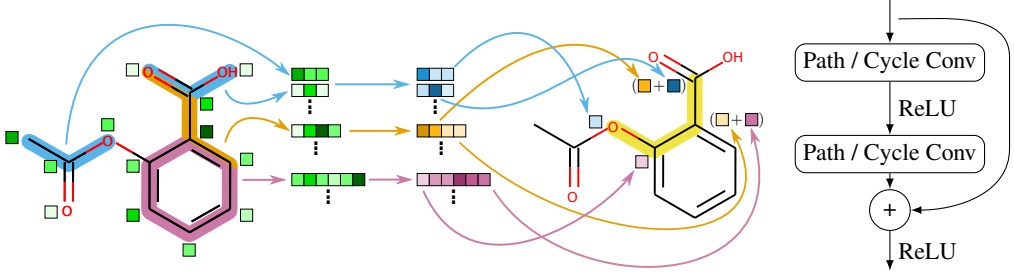

Figure 5: The internal structure of a single layer in the Autobahn architecture. We extract all path and cycle subgraphs of fixed length and their corresponding activations. (For compactness and readability, only some activations are shown.) We then apply a series of convolutional layers. A block diagram for this step is given on the right; each isomorphism class of reference domains has its own weights. Finally, we construct the activations for the next layer by narrowing and promoting between subgraphs and summing over the resulting promoted activations.

Autobahn can be performed "densely" using the $k$-th order network, and that choosing an Autobahn template that covers all sets of $k$ nodes recovers a $k$-th order network.

The strategy of using subgraphs to improve the expressiveness of a neural network is shared by the GSN network [21], which augments a message-passing neural network with node features that count the isomorphism classes of subgraphs a particular node is in. Consequently, it is reasonable to ask if using subgraphs for computation (as Autobahn does) gives any advantages compared to merely using them to create initial features. In Subsection 6.2 of the Supplement , we answer this in the affirmative. Moreover, while it was noted that a GSN network would be able to reconstruct a graph from its $n-1$ subgraphs if the reconstruction conjecture [44, 45] held, it is not clear if this task could be accomplished if the reconstruction conjecture was false. In contrast, we show that the high-order activations and the transfer of information using narrowing and promotion allows Autobahn to reconstruct a graph using neurons operating on its subgraphs of size $n-1$ independently of the reconstruction conjecture.

# 6   Molecular graphs on the Autobahn

With the Autobahn formalism defined, we return to our motivating task of learning the properties of molecular graphs. From the structure of organic molecules, we see that they often have a combination of a sparse chain-like "backbone" and cyclic structures such as aromatic rings. The importance of these structures is further justified by the theory of molecular resonance. Whereas in molecular graphs edges correspond to individual pairs of electrons, real electrons cannot be completely localized to single bonds. To re-inject this physics into graph representations of molecules, chemists construct "resonance structures": alternate molecular graphs formed by concertedly moving the electrons in a molecular graph. Importantly, the rules of chemical valency ensure that these motions occur almost exclusively on paths or cycles within the graph. In fact, cycle and path featurizations have already been used successfully in cheminformatic applications [46].

Motivated by these considerations, we will choose our local graphs to correspond to directed cycles and paths in graph. This has the additional advantage that the one-dimensional convolutions given by (6) are equivariant to the graph's automorphism group, and can be used directly. To construct the neurons for our architecture, we extract all paths of length three through six in the graph and all cycles of five or six elements. These cycle lengths were chosen because cycles of five and six elements are particularly common in chemical graphs. For each path or cycle, we construct two neurons corresponding to two ways of traversing the graph: for cycles, this corresponds to clockwise or anticlockwise rotation of the cycle, and for paths this sets one of the ends to be the "initial" node.

We then construct initial features for each neuron by embedding atom and bond identities as categorical variables. Embedded atom identities are then directly assigned to the corresponding points in each path or cycle. To assign the embedded bond identities, we arbitrarily assign each bond to the preceding node in the traversals mentioned above. Since we construct a neuron for both traversal directions, this procedure does not break permutation equivariance of the architecture. Following the

| Model | ZINC 10k (MAE, ↓) | ZINC full (MAE, ↓) | MolPCBA (AP ↑) | MolHIV (ROCAUC ↑) | MUV (AP ↑) |
|---|---|---|---|---|---|
| GCN | $0.367 \pm 0.011$ | N/A | $0.222 \pm 0.002$ | $0.788 \pm 0.080$ | N/A |
| GSN | $0.108 \pm 0.018$ | N/A | N/A | $0.780 \pm 0.010$ | N/A |
| DGN | $0.169 \pm 0.003$ | N/A | N/A | $\mathbf{0.797 \pm 0.010}$ | N/A |
| GINE-E | $0.252 \pm 0.015$ | $0.088 \pm 0.002$ | $0.227 \pm 0.003$ | $0.788 \pm 0.080$ | 0.091 |
| HIMP | $0.151 \pm 0.006$ | $0.032 \pm 0.002$ | $\mathbf{0.274 \pm 0.003}$ | $0.788 \pm 0.080$ | $0.114 \pm 0.041$ |
| **Ours** | $\mathbf{0.106 \pm 0.004}$ | $\mathbf{0.029 \pm 0.001}$ | 0.270 | $0.780 \pm 0.003$ | $\mathbf{0.119 \pm 0.005}$ |

Table 1: Performance of our Autobahn architecture on two splits of the ZINC dataset and three datasets in the OGB benchmark family, compared with other recent message passing architectures. ZINC experiments use MAE (lower is better); for all other metrics higher is better. Baselines were taken from [23, 47, 48] and [21].

initial featurization, we then construct layers using the four step procedure described in Section 5. The layer is illustrated in Figure 5.

Full details of the model, including training hyper-parameters and architecture details, are available in the Supplement; code is freely available at https://github.com/risilab/Autobahn. We present empirical results from an implementation of our architecture on two subsets of the ZINC dataset using the data splits from [23], as well as three standardized tasks from Open Graph Benchmark. All datasets are released under the MIT license. Baselines were taken from [21, 23, 47, 48]. Our automorphism-based neural network achieves results competitive with modern MPNNs.

# 7 Conclusion

In this paper, we have introduced Automorphism-based Neural Networks (Autobahn), a new framework for constructing neural networks on graphs. To build an Autobahn, we first choose a collection of template graphs. We then break our input graph into a collection of local graphs, each isomorphic to a template. Computation proceeds on each local graph by applying convolutions equivariant to the template's automorphism group, and by transferring information between the local graphs using two operators we refer to as "narrowing" and "promotion". MPNNs are specific examples of Autobahn networks constructed by choosing star-shaped templates applied to local neighborhoods. Similarly, applying Autobahn to a grid graph recovers steerable CNNs. Our experimental results show that Autobahn networks can be competitive with modern MPNNs on several molecular learning tasks.

We expect the choice of substructure to critically influence Autobahn performance. In future work we hope to explore the space of new models opened up by our theory. In learning situations where much is known about the graphs' structure, we believe practitioners will be able to choose templates that correspond to desired inductive biases, giving improved results. For instance, in future work we hope to improve our results on molecular graphs by adding templates that correspond to specific functional groups. For arbitrary graphs, it is not clear that the star graphs used by MPNNs are optimal or just a historical accident. It is possible that other "generic" templates exist that give reasonable results for a wide variety of graphs. For example, [2, 49] used path activations in conjunction with stochastic sampling strategies and the architecture in [50] can be viewed as using tree-like substructures. By further exploring the space of possible templates, we hope to construct richer graph neural networks that more naturally reflect the graphs on which they operate.

## 7.1 Broader Impacts

In our framework, the choice of template reflects practitioners' beliefs about which graph substructures are important for determining its properties. For social networks communities from different cultural backgrounds might result in graphs with differing topologies. Consequently, when applying Autobahn to these graphs care must be taken that chosen templates do not implicitly bias our networks towards specific cultural understandings.

# 8 Acknowledgements

This project was supported by DARPA "Physics of AI" grant number HR0011837139, and used computational resources acquired through NSF MRI 1828629. The Flatiron Institute is a division of the Simons Foundation. We thank Sonya Hanson, John Herr, Joe Paggi, and Helen Yu for useful feedback.

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
