# Supplementary Information for Autobahn: Automorphism-based Graph Neural Nets

## 1 Activations as functions on a group

In the Autobahn formalism, we make extensive use of the fact that the activations of a group-equivariant neural network can be treated as functions on the same group. Here we give a brief review for the unfamiliar reader. This formalism is also covered in detail in Sections 3 and 4 of Reference [6], although under slightly different conventions.

Consider a space $\mathcal{X}$ acted on by a group $G$: at every point $x$ in $\mathcal{X}$, we can apply a group element $g \in G$, which maps $x$ to another point in $\mathcal{X}$. The action of the group on $\mathcal{X}$ induces an action on functions of $\mathcal{X}$. We define an operator $T_g$ acting on functions $f : \mathcal{X} \to \mathbb{C}$ as follows. [1]

$$T_g\left(f\right)\left(x\right) = f\left(g\left(x\right)\right). \tag{1}$$

The inputs to group-equivariant neural networks are precisely functions on such spaces. For instance, for standard convolutional layers acting on images, each point on the space is a single pixel and the group of translation moves between pixels. The RGB value of each pixel is then vector-valued function of $\mathcal{X}$.

But while the input is a function on $\mathcal{X}$, representations internal to the network can be more general. For instance, consider a neural network that is given a list of objects and attempts to learn an adjacency matrix. Here, $\mathcal{X}$ is the list of objects, and the object identity is a function on the list. But the output space is a function on *pairs* of objects: a different space.

Fortunately, the complexity of dealing with a myriad of spaces can be avoided by mapping functions from their individual spaces to functions on $G$. This allows all possible spaces and group actions to be treated using a single (albeit abstract) formalism, simplifying definitions and proofs. For simplicity, we will assume that $G$ is transitive on $\mathcal{X}$: for all $x, y \in \mathcal{X}$, there exists a group element $g$ such that $g(x) = y$. (If the $G$ is not transitive, we can simply apply this procedure on every orbit of $G$ and concatenate the results along a channel dimension.) The construction proceeds as follows. We arbitrarily choose an initial point $x_0$ in $\mathcal{X}$ to act as the origin. Then, we construct the function

$$f_G\left(\sigma\right) = f\left(\sigma\left(x_0\right)\right) \quad \forall \sigma \in G. \tag{2}$$

Since we have assumed that $G$ is transitive, this is an injective map into functions on $G$ that preserves all of the information in $f$. In Figure 1, we depict this procedure, mapping the adjacency matrix for a graph with size 6 vertices to the $S_6$, the group of all permutations of six elements. Subsequent group actions then transform the resulting function as follows.

$$T_g f_G\left(\sigma\right) = f\left(\sigma(gx_0)\right) = f_G\left(\sigma g\right). \tag{3}$$

---

[1] Note this is a different convention for the action of group elements on functions from the one described in Reference [6]. The choice of whether to use $T_g$ operator described here or the $\mathbb{T}_g$ operator described in the reference is a matter of personal preference as they are inverses.

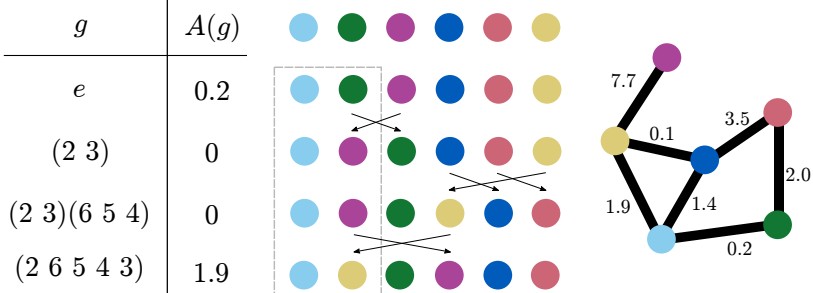

| $g$ | $A(g)$ |
|---|---|
| $e$ | 0.2 |
| (2 3) | 0 |
| (2 3)(6 5 4) | 0 |
| (2 6 5 4 3) | 1.9 |

Figure 1: Example showing how a function on the symmetric group can be constructed from neural network activations on the edges of a graph. To construct the function, we first list the vertices in arbitrary order. Then, for every permutation, we permute the indices and look at two arbitrarily chosen vertices (here we have chosen the first two). If these vertices form an edge, the function takes the edge activation as its value. Otherwise, the function takes a value of zero.

## 2 Equivariance of Automorphism-based Neurons

Here, we prove that the Algorithm 1 is equivariant to permutation.

**Theorem 1.** *Let $\mathcal{G}$ be a graph of $n$ vertices and let $\sigma \in \mathbb{S}_n$. The neuron $\mathfrak{n}^\ell$ described in Algorithm 1 obeys*

$$\mathfrak{n}^\ell\left(T_\sigma f^{\ell-1}\right) = T'_\sigma \mathfrak{n}^\ell\left(f^{\ell-1}\right) \tag{4}$$

*Proof.* Denote the permutation of $\mathcal{G}$ by $\sigma$ by $\bar{\mathcal{G}}$. Applying $\mathfrak{n}^\ell$ to $\bar{\mathcal{G}}$ constructs a matching $\bar{\mu}$. Since matching is accomplished up to an element in $\mathcal{T}$'s automorphism group, there exists $v \in \text{Aut}(\mathcal{T})$ such that

$$v\mu = \bar{\mu}\sigma$$

Denoting convolution over $\text{Aut}(\mathcal{T})$ as $*$, we have

$$\begin{aligned}
\mathfrak{n}^\ell\left(T_\sigma f^{\ell-1}\right) &= T'_{\bar{\mu}^{-1}}\left(\nu\left(\left(T_{\bar{\mu}}T_\sigma f^{\ell-1}\right)*w+b\right)\right) \\
&= T'_{\bar{\mu}^{-1}}\left(\nu\left(\left(T_v T_\mu f^{\ell-1}\right)*w+b\right)\right) \\
&= T'_{\bar{\mu}^{-1}}T'_v\left(\nu\left(\left(T_\mu f^{\ell-1}\right)*w+b\right)\right) \\
&= T'_{\sigma^{-1}}T'_{\mu^{-1}}\left(\nu\left(\left(T_\mu f^{\ell-1}\right)*w+b\right)\right).
\end{aligned}$$

which proves equivariance. Note the third line follows from equivariance of convolution over $\text{Aut}(\mathcal{T})$ and the fact that $b$ is invariant to elements in $\text{Aut}(\mathcal{T})$. □

## 3 Application of Autobahn to grid graphs

Here, we discuss the application of Autobahn to grid graphs and show how the ideas in Autobahn can be used to recover the standard convolutional and steerable CNN (p4m) architectures

### 3.1 Steerable CNNs

We first recover the steerable CNN architecture for the p4m group described in Reference [2]. For concreteness, we will consider a steerable CNN constructed using a $3 \times 3$ filter; however, the same construction can be applied to an arbitrary $k \times k$ filter. To recover a $3 \times 3$ steerable CNN, we will use as our template a grid of size $5 \times 5$. Using this larger template allows us to easily express the aggregation of all the signals in the



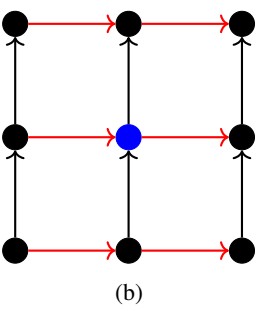

(a)

(b)

Figure 2: Template graphs for a Steerable CNN (a) and a standard convolutional neural network (b). In each case the output of the neuron is nonzero only for permutations that preserve the set of blue nodes and is invariant to permutation of the black nodes.

neuron's immediate receptive domain. Note that this resembles the use of a star graph for MPNNs where we include both the "input" and the "output" nodes in the same template.

We first describe the output of the neuron Each neuron outputs a function on $\mathbb{S}_{25}$ that (a) is nonzero only for permutations that preserve the center $4 \times 4$ nodes and (b) is constant for all permutations of the black nodes amongst each other. Furthermore, the output is nonzero only for elements corresponding to the automorphism group of the $3 \times 3$ grid. This is the group $D_4$ which corresponds to rotations of the grid by 90 degrees and horizontal / vertical reflections.

These outputs then form the input of the neuron in the next layer. Each neuron receives as input a function over $D_4$ associated with each $3 \times 3$ subgrid in Figure 2a. After narrowing and promotion, each of these function is embedded on group elements of $\mathbb{S}_{25}$ that first send the grid to the indices $1, \ldots, 9$ and then applies the permutation corresponding to the appropriate element of $D_4$. Finally, each neuron applies a convolution that combines subgrid group elements of similar orientations together using a convolution over the automorphism group of the template, which is also $D_4$.

This example highlights the importance of having a formalism capable of more complex methods of transferring information between neurons than merely copying over node or edge features. If the input features had been sent to individual nodes and edges than we would have inadvertently averaged input signals over nodes or edges shared between incoming $3 \times 3$ graphs.

## 3.2 Convolutional Neural Networks

As discussed in the main text, convolutional neural networks have a notion of left-right and up-down. This is information not present in a grid graph. To hope to recover a CNN from any graph neural network architecture, we must therefore consider a richer graph embedding for the image.

For a one-dimensional CNN, we can recover a CNN by considering a directed graph where edges always point in one direction. Two extend this to a two-dimensional CNN, we introduce two different types of edges: "red" edges that move horizontally from pixel to pixel in the image and "black" edges that move vertically. We arbitrarily set the red edges to always point towards the right and the black edges to always point up. Then, we construct a template using the same red and black edges (depicted in Figure 2b). Each template then picks out a single $k \times k$ subgrid in the image (again, we set $k = 3$ for specificity in the discussion that follows). To construct each neuron, we first extend our definition of automorphism to require that automorphism must also preserve color. In this case the automorphism group of the template is the trivial group and we can take an arbitrary linear combination of all the pixels covered by our template. This is the operation performed by a standard CNN filter.

This example gives insight into the role of the automorphism group in the Autobahn formalism. In the

main text the automorphism group was defined as the set of all permutations that leaves the Adjacency matrix unchanged. This choice was made simply because every graph has an adjacency matrix, ensuring that the automorphism group would be well-defined for any graph. However, if we know more about the structure of our graph, there is no reason that information cannot be included into the definition. For instance, in this example we have also required that the permutations leave a collection of edge features unchanged as well (specifically edge color). In other applications one might also further constrain the group by considering other graph attributes such as node features.

# 4    Independence of representative for promotion.

Uniqueness of $u$ and $s$ in the definition of promotion is a straightforward application of the following lemma.

**Lemma 2.** *Let $\tau$ and $t$ be elements of $\mathbb{S}_m$. Let $u, v \in \mathbb{S}_k$ and $s, q \in \mathbb{S}_{m-k}$ be permutations such that*

$$\tau = \acute{u}\grave{s}t = \acute{v}\grave{q}t \tag{5}$$

*Then $u = v$ and $s = q$.*

*Proof.* It follows directly from the assumption that

$$\acute{u}\grave{s} = \acute{v}\grave{q}$$
$$\implies \acute{u}\grave{s}\acute{v}^{-1}\grave{q}^{-1} = \mathcal{I}$$

Since $\acute{v}^{-1}$ acts only on the first $k$ elements and $\grave{s}$ acts only on the last $k$, they commute and

$$\acute{u}\acute{v}^{-1}\grave{s}\grave{q}^{-1} = \mathcal{I}$$

Moreover, let $a = uv^{-1}$ and $b = sq^{-1}$. It follows from the definition of ´ and ` that

$$\acute{a} = \acute{u}\acute{v}^{-1} \text{ and } \grave{b} = \grave{s}\grave{q}^{-1}. \tag{6}$$

implying

$$\acute{a}\grave{b} = \mathcal{I}. \tag{7}$$

But this is only possible if $a$ is the identity element of $\mathbb{S}_k$ and $b$ is the identity element of $\mathbb{S}_{m-k}$, which in turn implies $u = v$ nd $s = q$. $\qquad\square$

# 5    Proof of Equivariance for Autobahn

Here, we prove that the Autobahn architecture obeys permutation equivariance. For Autobahn, neural network inputs and activations are functions on subgroups acting on a subset of the graph's vertices and global permutation of inputs can induce group operations in the associated subgroups.

Proving equivariance requires we describe how Autobahn networks transform when applying a global permutation to the input graph. To do so, we recall from Subsection 4.2 that narrowing is the pseudoinverse of promotion. Consequently, between every step in Autobahn we can promote the activation of any neuron to $\mathbb{S}_n$ and then immediately narrow it back to the nodes in the neuron's local graph without changing the network. Doing this allows us to rewrite the steps for an Autobahn neuron $\mathfrak{n}_j^\ell$ as follows.

T1.1   Narrow every activation $f_{s_z}^{\ell-1}$ from $S_n$ to $\left(a_1^z, \ldots, a_{m_z}^z\right)$.

T1.2   Further narrow the incoming activation $f_{s_z}^{\ell-1}$ to the corresponding intersection to get $f_{s_z}^{\ell-1}\!\downarrow_{(b_1^z \ldots b_{k_z}^z)}$.

T1.3   Promote each $f_{s_z}^{\ell-1}\!\downarrow_{(b_1^z \ldots b_{k_z}^z)}$ to $S_n$.

T2.1   Narrow the results back to $\left(b_1^z \ldots b_{k_z}^z\right)$.

T2.2  Promote each of these to $(a_1, \ldots, a_m)$:

$$\tilde{f}_z = f_{s_z}^{\ell-1} \downarrow_{(b_1^z \ldots b_{k_z}^z)} \uparrow^{(a_1 \ldots a_m)}.$$

T2.3  Promote $\tilde{f}_z$ to $S_n$.

T3.1  Narrow back down to $(a_1, \ldots, a_m)$.

T3.2  Apply a symmetric polynomial $S$ to $\tilde{f}_1, \ldots, \tilde{f}_p$:

$$\widehat{f} = S(\tilde{f}_1, \ldots, \tilde{f}_p).$$

T3.3  Promote $\widehat{f}$ to $S_n$.

T4.1  Narrow back down to $(a_1, \ldots, a_m)$.

T4.2  Apply one or more (learnable) equivariant linear transformations $p_j$, each of which is followed by a fixed pointwise nonlinearity $\xi$, to get the final output of the neuron.

$$f_j^\ell = \xi(p_j(\widehat{f})).$$

T4.3  Promote every $f_j^\ell$ to $S_n$.

Here, we have rewritten Autobahn so that steps T1, T2, T3, and T4 all map functions on $\mathbb{S}_n$ to functions on $\mathbb{S}_n$. Consequently, we can show that each of these steps is equivariant to permutations in $\mathbb{S}_n$, making the network to equivariant to permutation as whole.

## 5.1  Action of permutations on narrowing and promotion

Before we prove equivariance, we first consider how a permutation of a local graphs affects the arbitrary orderings chosen when narrowing and promoting. Consider narrowing a function on $m$ vertices onto a subset of $k$ vertices, $\{i_1, \ldots, i_k\} \subset \{1, \ldots, m\}$. Narrowing first applies a permutation that sends the vertices indexed by $\{i_1, \ldots, i_m\}$ to the first $k$ indices, $(1, \ldots, m)$. Vertex $i_1$ is sent to position 1, vertex $i_2$ is sent to position 2, and so forth. However, this implicitly orders the vertices: If we had initially listed the vertices in a different order then they would have been sent to different positions in the set $(1 \ldots, m)$. Consequently, when applying a permutation $\pi$ to a local graph graph with $m$ nodes, it is not enough to merely consider narrowing from $\{\pi(i_1), \ldots, \pi(i_m)\}$ to $(1, \ldots, m)$ as we have no guarantee of recovering the same arbitrary ordering when considering a permuted copy of the graph. Rather, we must also ensure that our network is unaffected by subsequent permutation $p$ of the labels $(1, \ldots, m)$ caused by making a different arbitrary choice in order for the permuted graph. Throughout this section, we will extend the ´ and ` notation to both maps from $\mathbb{S}_m$ to $\mathbb{S}_n$ or $\mathbb{S}_k$ to $\mathbb{S}_n$ as necessary and trust that the precise domain and ranges of the maps will be clear from context.

Let $f$ be an activation and $u \in \mathbb{S}_k$, $s \in \mathbb{S}_{m-k}$, and $t \in \mathbb{S}_m$ be permutations as in Subsection 5.2 of the main text. Upon applying a permutation $\pi$ to a neuron's local graph, the neuron narrows the permuted activation $T_\pi f$ to the function

$$(T_\pi)f\downarrow_{(\pi(i_{p(1)}), \pi(i_{p(2)}), \ldots, \pi(i_{p(k)}))}(u) = (n-k)!^{-1} \sum_{s \in \mathbb{S}_{m-k}} T_\pi f(\acute{u}\grave{s}\tau). \tag{8}$$

where $\tau$ is an arbitrary permutation that sends $\pi(i_1)$ to $p(1)$, $\pi(i_2)$ to $p(2)$, etc. Moreover, since $t$ and $\tau\pi$ send the same vertices to the (unordered) sets $\{1, \ldots, k\}$ and $\{k+1, \ldots, m\}$ there must exist a permutation $a \in \mathbb{S}_{k-m}$ such that

$$\acute{p}\grave{a}t = \tau\pi. \tag{9}$$

$$(\ldots, 5, 3, \ldots, 2, 1, \ldots, 4 \ldots) \underset{b^{-1}}{\overset{b}{\rightleftarrows}} (1, 2, 3, 4, 5, \ldots) \underset{\acute{t}^{-1}}{\overset{\acute{t}}{\rightleftarrows}} (2, 4, 3, \ldots)$$

$$\rho^{-1} \Big\uparrow\Big\downarrow \rho \qquad\qquad \acute{\pi}^{-1}\grave{\alpha}^{-1} \Big\uparrow\Big\downarrow \acute{\pi}\grave{\alpha} \qquad\qquad \acute{p}^{-1}\acute{\grave{a}}^{-1} \Big\uparrow\Big\downarrow \acute{p}\grave{\acute{a}}$$

$$(\ldots, 4, 3, \ldots, 5, \ldots 2, 4, 1) \underset{\pi^{-1}}{\overset{\pi}{\rightleftarrows}} (5, 4, 1, 3, 2, \ldots) \underset{\tau^{-1}}{\overset{\tau}{\rightleftarrows}} (4, 2, 3, \ldots,)$$

Figure 3: Commutative diagram depicting the permutations involved in T1 and T2. For concreteness, we have set $k = 3$ and $m = 5$. Our local graph is defined on vertices vertices 1 through 5, and here we are narrowing to vertices $j_1 = 2$, $j_2 = 4$, and $j_3 = 2$.

and we can consequently write

$$
\begin{aligned}
T_\pi f\!\downarrow_{(\pi(i_{p(1)}), \pi(i_{p(2)}), \ldots, \pi(i_{p(k)}))}(u) &= (n-k)!^{-1} \sum_{s \in \mathbb{S}_{m-k}} T_\pi f(\acute{u}\grave{s}\tau). \\
&= (n-k)!^{-1} \sum_{s \in \mathbb{S}_{m-k}} T_\pi f(\acute{u}\grave{s}\acute{p}\grave{a}t\pi^{-1}). \\
&= (n-k)!^{-1} \sum_{s \in \mathbb{S}_{m-k}} f(\acute{u}\grave{s}\acute{p}\grave{a}t\pi^{-1}\pi). \\
&= \sum_{s \in \mathbb{S}_{m-k}} f(\acute{u}\acute{p}\grave{s}\grave{a}t). \\
&= f\!\downarrow_{(\pi(i_{p(1)}), \pi(i_{p(2)}), \ldots, \pi(i_{p(k)}))}(up) \\
&= T_p f\!\downarrow_{(\pi(i_{p(1)}), \pi(i_{p(2)}), \ldots, \pi(i_{p(k)}))}(u). \qquad (10)
\end{aligned}
$$

Similarly, promoted functions transform as

$$
g\!\uparrow^{(\pi(i_{p(1)}), \ldots, \pi(i_{p(k)}))}(\alpha) = \begin{cases} g(u) & \text{if } \exists u \in \mathbb{S}_k, s \in \mathbb{S}_{m-k} \text{ s.t. } \alpha = \acute{u}\grave{s}\tau \\ 0 & \text{otherwise.} \end{cases} \qquad (11)
$$

$$
= \begin{cases} g(u) & \text{if } \exists u \in \mathbb{S}_k, s \in \mathbb{S}_{m-k} \text{ s.t. } \alpha = \acute{u}\acute{p}\grave{s}\grave{a}t\pi^{-1} \\ 0 & \text{otherwise.} \end{cases} \qquad (12)
$$

$$
\implies T_p g\!\uparrow^{(\pi(i_{p(1)}), \ldots, \pi(i_{p(k)}))} = (T_\pi g)\!\uparrow^{(i_1, \ldots, i_k)} \qquad (13)
$$

## 5.2 Proof of equivariance for individual sublayers

We now prove that each of the individual sublayers T1-T4 obey equivariance.

### 5.2.1 T1 is equivariant

Since we are applying narrowing twice, we must deal with two sets of ordered vertices: $(i_1, \ldots, i_m)$ and the ordered subset to which we are narrowing, $(j_1, \ldots, j_k)$.

Our entire graph will be acted on by a permutation $\rho$. We let $b \in \mathbb{S}_n$ be an arbitrary permutation that sends the ordered set $(i_1, \ldots, i_m)$ to the first $m$ positions and let $\beta \in \mathbb{S}_n$ be a permutation that sends $(\rho(i_1), \ldots, \rho(i_m))$ to $(\pi(1), \ldots, \pi(m))$ for some (unknown) $\pi$. The permutations $b$ and $\beta$ play the same role as $t$ and $\tau$ when narrowing from $\{1, \ldots, n\}$ to $(i_1, \ldots, i_m)$.

To aid the reader, we have summarized how global permutations affect the various subsets involved in narrowing and promotion in 3.

We now seek to prove equivariance:

$$\left(\left(\left((T_\rho f)\downarrow_{\left(\rho\left(i_{\pi(1)}\right),...,\rho\left(i_{\pi(m)}\right)\right)}\right)\downarrow_{\left(\pi\left(j_{p(1)}\right),...,\pi\left(j_{p(k)}\right)\right)}\right)\uparrow^{\left(\rho\left(i_{j_{p(1)}}\right)...\rho\left(i_{j_{p(k)}}\right)\right)}\right.$$
$$= T_\rho\left(\left(\left(f\downarrow_{(i_1,...,i_m)}\right)\downarrow_{(j_1,...,j_k)}\right)\uparrow^{\left(i_{j_1},...,i_{j_k}\right)}\right) \tag{14}$$

Repeatedly applying (10), we have

$$\left(\left(\left((T_\rho f)\downarrow_{\left(\rho\left(i_{\pi(1)}\right),...,\rho\left(i_{\pi(m)}\right)\right)}\right)\downarrow_{\left(\pi\left(j_{p(1)}\right),...,\pi\left(j_{p(k)}\right)\right)}\right)\uparrow^{\left(\rho\left(i_{j_{p(1)}}\right)...\rho\left(i_{j_{p(k)}}\right)\right)}\right.$$
$$= \left(\left(T_\pi\left(f\downarrow_{(i_1,...,i_m)}\right)\right)\downarrow_{\left(\pi\left(j_{p(1)}\right),...,\pi\left(j_{p(k)}\right)\right)}\right)\uparrow^{\left(\rho\left(i_{j_{p(1)}}\right)...\rho\left(i_{j_{p(k)}}\right)\right)} \tag{15}$$
$$= \left(T_p\left(\left(f\downarrow_{(i_1,...,i_m)}\right)\downarrow_{(j_1,...,j_k)}\right)\right)\uparrow^{\left(\rho\left(i_{j_{p(1)}}\right)...\rho\left(i_{j_{p(k)}}\right)\right)} \tag{16}$$
$$= T_\rho\left(\left(\left(f\downarrow_{(i_1,...,i_m)}\right)\downarrow_{(j_1,...,j_k)}\right)\right)\uparrow^{\left(i_{j_1}...i_{j_k}\right)} \tag{17}$$

where the last line follows by the same argument as (13).

### 5.2.2 T2 is equivariant

We seek to prove that

$$\left(\left(\left((T_\rho f)\downarrow_{\left(\rho\left(i_{j_{p^{-1}(1)}}\right)...\rho\left(i_{j_{p^{-1}(k)}}\right)\right)}\right)\uparrow^{\left(\pi\left(j_{p(1)}\right),...,\pi\left(j_{p(k)}\right)\right)}\right)\uparrow^{\left(\rho\left(i_{(1)}\right),...,\rho\left(i_{(m)}\right)\right)}\right.$$
$$= T_\rho\left(\left(\left(f\downarrow_{\left(i_{j_1},...,i_{j_k}\right)}\right)\uparrow^{(j_1,...,j_k)}\right)\uparrow^{(i_1,...,i_m)}\right) \tag{18}$$

The proof proceeds similarly to the proof for T1. We have

$$\left(\left(\left((T_\rho f)\downarrow_{\left(\rho\left(i_{j_{p(1)}}\right)...\rho\left(i_{j_{p(k)}}\right)\right)}\right)\uparrow^{\left(\pi\left(j_{p(1)}\right),...,\pi\left(j_{p(k)}\right)\right)}\right)\uparrow^{\left(\rho\left(i_{\pi(1)}\right),...,\rho\left(i_{\pi(m)}\right)\right)}\right.$$
$$= \left(\left(T_p\left(f\downarrow_{\left(i_{j_1},...,i_{j_k}\right)}\right)\right)\uparrow^{\left(\pi\left(j_{p(1)}\right),...,\pi\left(j_{p(k)}\right)\right)}\right)\uparrow^{\left(\rho\left(i_{\pi(1)}\right),...,\rho\left(i_{\pi(m)}\right)\right)} \tag{19}$$
$$= \left(T_\pi\left(\left(f\downarrow_{\left(i_{j_1},...,i_{j_k}\right)}\right)\uparrow^{(j_1,...,j_k)}\right)\right)\uparrow^{\left(\rho\left(i_{\pi(1)}\right),...,\rho\left(i_{\pi(m)}\right)\right)} \tag{20}$$
$$= T_\rho\left(\left(\left(f\downarrow_{\left(i_{j_1},...,i_{j_k}\right)}\right)\uparrow^{(j_1,...,j_k)}\right)\uparrow^{(i_1,...,i_m)}\right) \tag{21}$$

### 5.2.3 T3 is equivariant

To prove that applying a symmetric polynomial and applying convolution over a graph's automorphism group preserves equivariance, we require the following lemma.

**Lemma 3.** *Let A be an $\mathbb{S}_m$ equivariant operator. Then,*

$$\left(A\left((T_\rho f)\downarrow_{\left(\rho(i_{\pi(1)}),...,\rho(i_{\pi(k)})\right)}\right)\right)\uparrow^{\left(\rho(i_{\pi(1)}),...,\rho(i_{\pi(k)})\right)} = T_\rho f\left(\left(A\left(f\downarrow_{(i_1,...,i_k)}\right)\right)\uparrow^{(i_1,...,i_k)}\right) \tag{22}$$

*Proof.* Applying (10), the definition of equivariance, and (13) we have

$$\left(A\left((T_\rho f)\downarrow_{\left(\rho(i_{\pi(1)}),...,\rho(i_{\pi(k)})\right)}\right)\right)\uparrow^{\left(\rho(i_{\pi(1)}),...,\rho(i_{\pi(k)})\right)}$$
$$= \left(A\left(T_\pi\left(f\downarrow_{i_1,...,i_m}\right)\right)\right)\uparrow^{\left(\rho(i_{\pi(1)}),...,\rho(i_{\pi(k)})\right)}$$
$$= \left(T_\pi\left(A\left(f\downarrow_{i_1,...,i_m}\right)\right)\right)\uparrow^{\left(\rho(i_{\pi(1)}),...,\rho(i_{\pi(k)})\right)}$$
$$= T_\rho f\left(\left(A\left(f\downarrow_{(i_1,...,i_k)}\right)\right)\uparrow^{(i_1,...,i_k)}\right) \tag{23}$$

$\square$

To prove equivariance of T3, it is therefore enough to prove that application of the symmetric polynomial is $\mathbb{S}_m$-equivariant. However, the output of the polynomial is invariant by definition, and it is well-known that the products are group-equivariant [5]. Consequently, the symmetric polynomials obey equivariance.

### 5.2.4 T4 is equivariant

Equivariance of T4 follows directly from Lemma 3 and the fact the neuron described by Algorithm 1 is equivariant, as shown in Section 2.

# 6 Expressivity of Autobahn

Here, we give results directly comparing the expressivity of specific Autobahn architectures to other graph neural networks.

## 6.1 Recovery of k'th-order Networks

Here, we show that

1. The expressivity of an Autobahn network whose largest template has $k$ nodes is bounded above by the expressivity of a $k$'th order network as described in [7].

2. There exists an Autobahn network that achieves this bound.

### 6.1.1 Group-theoretic description of k'th order networks

To facilitate our discussion, we first describe the $k$'th order networks from a group theoretic point of view. Here, the $k$'th order network can be described as functions on the quotient spaces $\mathbb{S}_n/\mathbb{S}_{n-k}$, $\mathbb{S}_n/\mathbb{S}_{n-k+1}$, etc. The different orbits of the permutation group on the $k$'th order tensor correspond to different quotient spaces: for instance, the central diagonal Note that in general, a function on $\mathbb{S}_n/\mathbb{S}_{n-m}$ can be identified with a function on $\mathbb{S}_n$ that obeys the following property

$$g(u) = g(u\grave{s}) \ \forall \ s \in \mathbb{S}_{n-m}. \tag{24}$$

The value of such a function depends only on which nodes $u$ sends to the first $m$ positions: the indices of these nodes correspond, in order, to the indices in a tensor representation of the data. Note that the original formulation of the $k$'th order network is given in terms of tensors with indices corresponding to individual nodes; the two formulations can be interconverted as described in Section 1. The action of the network itself can be fully described using the general theory of group equivariant networks described in [6]. However, here it will be easier to describe the network as the composition of three operations.

1. Convolution over all possible permutations of the first $m$ indices

$$g \star h(u) = \sum_{s \in \mathbb{S}_k} g(\acute{s}u)h(s^{-1}) \tag{25}$$

Here $g$ is a function obeying (24) and $h$ is a function from $\mathbb{S}_k \to \mathcal{C}$. It is easy to see that the output of the convolution is on the same quotient space as $g$. Note that by setting $h$ to a function that is 1 for a single group element and zero otherwise and convolving, it is possible to use (25) to translate $g$ by a permutation that permutes the first $k$.

2. Moving a function onto a smaller homogeneous space by averaging over all possible permutations of the other indices. If we look at the corresponding functions on $\mathbb{S}_n$, this corresponds to performing the following sum.

$$g'(u) = \sum_{s \in \mathbb{S}_{n-k}} g(\acute{s}u) \tag{26}$$

This corresponds to averaging over the last $n - k$ indices.

3. Moving a function from a smaller homogeneous space $\mathbb{S}_n/\mathbb{S}_{n-m}$ to a larger homogeneous space $\mathbb{S}_n/\mathbb{S}_{n-k}$. The precise characterization of this function depends on how we view the inputs and outputs. Writing them as $m$'th order and $k$'th order tensors $\gamma$ and $\gamma'$ respectively, this corresponds to setting

$$\gamma'_{i_1, i_2, \ldots, i_m, \ldots, i_k} = \gamma_{i_1, i_2, \ldots, i_m}. \tag{27}$$

However, viewed as a function on $\mathbb{S}_n$, this leaves the function unchanged.

### 6.1.2 Embedding Autobahn Activations in a k'th order network

It is sufficient to prove that the $k$'th order network can reproduce Autobahn for templates of size $k$ and a $k$'th order signal, as we can trivially treat a lower-order signal as a $k$'th order signal that has the same value for many choices of index. First, we show that the $k$'th order network is capable of representing the Autobahn activation. Consider an Autobahn activation $f_j$ whose local graph is denoted $\mathfrak{R}_j$. Using promotion, we can embed the activation as a function on $\mathbb{S}_n$ as follows. Let $t$ be an arbitrary permutation that moves the indices of the $j$'th local graph to $1, \ldots, k$.

$$f_j \uparrow^{(1, \ldots, n)}(\alpha) = \begin{cases} f_j(u) & \text{if } \exists u \in \mathbb{S}_k, s \in \mathbb{S}_{m-k} \text{ s.t. } \alpha = \acute{u}\grave{s}t \\ 0 & \text{otherwise.} \end{cases} \tag{28}$$

More specifically, we note that since the general theory of group equivariant networks says the activation of an Autobahn neuron (here arbitrarily chosen to be neuron $j$) can be written as a function over $\mathrm{Aut}(\mathfrak{R}_j)$, and will consequently have an even more structured sparsity pattern. Specifically, we define the $\hat{\ }$ symbol, similar to the $\grave{\ }$, that maps elements of $\mathrm{Aut}(\mathfrak{R}_j)$ to $\mathbb{S}_n$ such that for $u \in \mathrm{Aut}(\mathfrak{R}_j)$,

$$\hat{u}(i) = u(i) \quad \forall\, 1 \le i \le k \tag{29}$$
$$\hat{u}(i) = i \quad \forall\, i > k. \tag{30}$$

Note that if a permutation can be written as $\hat{u}$, it can also be written using the $\acute{\ }$ symbol using Cayley's theorem. Consequently, for any given choice of $t_j$ that sends the indices here exists a permutation in $v_j \in \mathbb{S}_k$ such that

$$f_j \uparrow^{(1, \ldots, n)}(a) = \begin{cases} f_j(u) & \text{if } \exists u \in \mathrm{Aut}(\mathfrak{R}_j), s \in \mathbb{S}_{m-k} \text{ s.t. } a = \hat{u}\acute{v_j}\grave{s}t_j \\ 0 & \text{otherwise.} \end{cases} \tag{31}$$

Note this embedding is (trivially) injective.

We now extend this embedding to the multiple neuron case. Without loss of generality, we assume that each subgraph isomorphic to a template corresponds to single neuron. (If two neurons are operating on the same subgraph they can be treated as a single neuron operating on more channels.) We can always sort the neurons based on their isomorphism class without breaking permutation equivariance. Consequently we need only consider neurons that share a template when constructing an embedding: neurons whose local graphs are not isomorphic can be embedded separately, and then concatenated along the channel index. Denoting the set of neurons that share the template by $N_{\mathcal{T}}$, we can embed their activations in $\mathbb{S}_n/\mathbb{S}_{n-k}$ as

$$f^{emb}(a) = \sum_{j=1}^{N_{\mathcal{T}}} f_j \uparrow^{(1, \ldots, n)}(a) \tag{32}$$

Since each subgraph isomorphic to a template corresponds to a single neuron, for every value of (a) there is at most one promoted activation that is nonzero. Consequently, this embedding is also injective.

Finally, we note that Autobahn explicitly uses information about the location of subgraphs in the networks. This can be transferred into a $k$'th order network by embedding functions that are 1 on a neuron, or on the intersection between neurons, in the same manner as in (32).

### 6.1.3 Performing Automorphic Convolution in a k'th order network

We now show that the $k$'th order network is capable of performing operations on the embedded signal that correspond to the core operations of Autobahn: convolution over the Automorphism group, narrowing, and promotion. Let $f_j$ and $g$ be a functions on $\mathbb{S}_k$ that are nonzero only for elements of the Automorphism group of $\mathfrak{R}_j$. We first observe that

$$(f_j * g)\uparrow^{(1,\ldots,n)}(a) = f_j\uparrow^{(1,\ldots,n)} \star g(a) \tag{33}$$

To show this, we note that if the top condition in (31) holds, we have that

$$(f_j * g)\uparrow^{(1,\ldots,n)}(a) = \sum_{w\in\text{Aut}(\mathfrak{R}_j)} f_j(w^{-1}u)g(w) \tag{34}$$

$$= \sum_{w\in\text{Aut}(\mathfrak{R}_j)} f_j\uparrow^{(1,\ldots,n)}(\hat{w}^{-1}\hat{u}\acute{v}_j\grave{s}t_j)g(w) \tag{35}$$

$$= \sum_{w\in\mathbb{S}_k} f_j\uparrow^{(1,\ldots,n)}(\acute{w}^{-1}\hat{u}\acute{v}_j\grave{s}t_j)g(w) \tag{36}$$

$$= f_j\uparrow^{(1,\ldots,n)} \star g(a) \tag{37}$$

Similarly, if the bottom condition holds then

$$(f_j * g)\uparrow^{(1,\ldots,n)}(a) = 0 \tag{38}$$

$$= \sum_{s\in\mathbb{S}_k} f_j\uparrow^{(1,\ldots,n)}(\acute{s}a)g(s^{-1}) \tag{39}$$

$$= f_j\uparrow^{(1,\ldots,n)} \star g(a). \tag{40}$$

The second line follows from the fact that if $f_j(\acute{s}a)$ being nonzero implies that the first condition in (31) holds instead of the second, contradicting our assumption.

We now return to our embedding of the Autobahn activations. Embedding the output of each neurons convolution over their respective local graph's automorphism group,

$$\sum_{j=1}^{N_\mathcal{T}} (f * g)_j\uparrow^{(1,\ldots,n)}(a) = \sum_{j=1}^{N_\mathcal{T}} \sum_{s\in\mathbb{S}_k} f_j\uparrow^{(1,\ldots,n)}(\acute{s}a)g(s^{-1}) \tag{41}$$

$$= \left(\sum_{j=1}^{N_\mathcal{T}} \sum_{s\in\mathbb{S}_k} f_j\uparrow^{(1,\ldots,n)}(\acute{s}a)\right)g(s^{-1}) \tag{42}$$

$$= \left(\sum_{j=1}^{N_\mathcal{T}} f_j\uparrow^{(1,\ldots,n)}\right) \star g(a) \tag{43}$$

Consequently, convolutions over the automorphism groups of individual neurons can be written using convolutions of the form of (25) on the embedded signal.

### 6.1.4 Performing Narrowing in a k'th order network

We first establish two useful identities. We first note that

$$g\!\downarrow_{(i_1,\ldots,i_k)}\!\downarrow_{(j_1,\ldots,j_m)} = g\!\downarrow_{(i_{j_1},\ldots i_{j_m})} \tag{44}$$

and

$$g\!\downarrow_{(i_1,\ldots i_k)}\uparrow^{(1,\ldots,n)}(a) = \begin{cases} \sum_{w\in S_{n-m}} g(\acute{u}\grave{w}t) & \text{if } \exists u\in\mathbb{S}_k, s\in\mathbb{S}_{n-k} \text{ s.t. } a = \acute{u}\grave{s}t \\ 0 & \text{otherwise.} \end{cases} \tag{45}$$

This latter identity can be written more compactly by introducing an indicator function $\mathbb{1}$ that is 1 if the condition holds and 0 otherwise. Since

$$\exists u \in \mathbb{S}_k, s \in \mathbb{S}_{n-k} \text{ s.t. } a = \acute{u}\grave{s}t \iff \{a(i_1), \ldots, a(i_k)\} = \{1, \ldots, k\} \tag{46}$$

we can write

$$g\!\downarrow_{(i_1,\ldots i_k)}\!\uparrow^{(1,\ldots,n)}(a) = \left(\sum_{w \in S_{n-m}} g(\grave{w}a)\right)\mathbb{1}_{A_{\{i\}}}(a) \tag{47}$$

where

$$A_{\{i\}} = \{\alpha \in \mathbb{S}_n | \alpha(i_1), \ldots, \alpha(i_k)\} = \{1, \ldots, k\}\,. \tag{48}$$

This set is preserved by permutation of the other $N - k$ indices, and consequently

$$g\!\downarrow_{(i_1,\ldots i_k)}\!\uparrow^{(1,\ldots,n)}(a) = \sum_{w \in S_{n-m}} g(\grave{w}a)\mathbb{1}_{A_{\{i\}}}(\grave{w}a) \tag{49}$$

Combining (44) and (49), we have

$$\left(\sum_{i=1}^{N_{\mathcal{T}}} f_i\!\downarrow_{(j_1,\ldots,j_m)}\right)\uparrow^{(1,\ldots,n)}(a) = \sum_{i=1}^{N_{\mathcal{T}}} f_i\!\downarrow_{(j_1,\ldots,j_m)}\!\uparrow^{(1,\ldots,n)}(a) \tag{50}$$

$$= \sum_{i=1}^{N_{\mathcal{T}}} f_i\!\uparrow^{(1,\ldots,n)}\!\downarrow_{(i_1,\ldots,i_k)}\!\downarrow_{(j_1,\ldots,j_m)}\!\uparrow^{(1,\ldots,n)}(a) \tag{51}$$

$$= \sum_{i=1}^{N_{\mathcal{T}}} f_i\!\uparrow^{(1,\ldots,n)}\!\downarrow_{(i_{j_1},\ldots,i_{j_m})}\!\uparrow^{(1,\ldots,n)}(a) \tag{52}$$

$$= \sum_{i=1}^{N_{\mathcal{T}}} \sum_{s \in \mathbb{S}_{n-m}} f_i\!\uparrow^{(1,\ldots,n)}(\grave{s}a)\mathbb{1}_{B_j}(\grave{s}a) \tag{53}$$

where is the set of permutations such that

$$B_j = \{\alpha \in \mathbb{S}_n | \alpha(i_{j_1}), \ldots, \alpha(i_{j_m})\} = \{1, \ldots, m\}\,. \tag{54}$$

Here the $i$ indices are chosen from an arbitrary neuron containing the graph intersections; there is no dependence on $i$ on the left-hand side because the actual values of $i_{j_1}$ are independent of the neuron used. Summing both sides over the $M_{\mathcal{T}}$ possible intersections to which we are narrowing our activations gives

$$\sum_{j=1}^{M_{\mathcal{T}}}\left(\sum_{i=1}^{N_{\mathcal{T}}} f_i\!\downarrow_{(j_1,\ldots,j_m)}\right)\uparrow^{(1,\ldots,n)}(a) = \sum_{j=1}^{M_{\mathcal{T}}}\sum_{i=1}^{N_{\mathcal{T}}}\sum_{s \in \mathbb{S}_{n-m}} f_i\!\uparrow^{(1,\ldots,n)}(\grave{s}a)\mathbb{1}_{B_j}(\grave{s}a) \tag{55}$$

$$= \sum_{s \in \mathbb{S}_{n-m}}\left(\sum_{i=1}^{N_{\mathcal{T}}} f_i\!\uparrow^{(1,\ldots,n)}(\grave{s}a)\right)\left(\sum_{j=1}^{M_{\mathcal{T}}}\mathbb{1}_{B_j}(\grave{s}a)\right) \tag{56}$$

Consequently, we can write narrowing as an operation on a $k$'th order network by first applying an element-wise multiplication against a signal, followed by an average over the remaining indices.

### 6.1.5 Performing Promotion in a k'th order network

To write promotion as an operation in a $k$'th order network, let $\{f_j\}$ a function being promoted from a subgraph on indices $i_{j_1}, \ldots, i_{j_m}$ to $i_1, \ldots, i_k$. We observe that

$$g\!\downarrow_{(i_{j_1},\ldots,i_{j_m})}\!\uparrow^{(i_1,\ldots,i_k)}\!\uparrow^{(1,\ldots,n)}(a) = \sum_{w \in S_{n-m}} g(\grave{w}a)\mathbb{1}_{A_{\{i\}}}(a)\mathbb{1}_{B_j}(a). \tag{57}$$

Now, we consider all $M_{\mathcal{T}}$ functions being promoted from a set of isomorphic subgraphs. We have

$$\sum_{j=1}^{M_{\mathcal{T}}} f_j \uparrow^{(i_1,\ldots,i_k)} \uparrow^{(1,\ldots,n)}(a) = \sum_{j=1}^{M_{\mathcal{T}}} f_j \uparrow^{(1,\ldots,n)}(a) \downarrow^{(i_{j_1},\ldots,i_{j_m})} \uparrow^{(i_1,\ldots,i_k)} \uparrow^{(1,\ldots,n)} \tag{58}$$

$$= \sum_{j=1}^{M_{\mathcal{T}}} f_j \uparrow^{(1,\ldots,n)}(a) \mathbb{1}_{A_{\{i\}}}(a) \mathbb{1}_{B_j}(a) \tag{59}$$

Summing over the target reference domain gives

$$\sum_{i=1}^{N_{\mathcal{T}}} \sum_{j=1}^{M_{\mathcal{T}}} f_j \uparrow^{(i_1,\ldots,i_k)} \uparrow^{(1,\ldots,n)}(a) = \sum_{i=1}^{N_{\mathcal{T}}} \sum_{j=1}^{M_{\mathcal{T}}} f_j \uparrow^{(1,\ldots,n)}(a) \mathbb{1}_{A_{\{i\}}}(a) \mathbb{1}_{B_j}(a) \tag{60}$$

Finally, we note that since each element of $f_j \uparrow^{(1,\ldots,n)}(a)$ is nonzero only for a single $\mathbb{1}_{B_j}$, we can write

$$\sum_{i=1}^{N_{\mathcal{T}}} \sum_{j=1}^{M_{\mathcal{T}}} f_j \uparrow^{(i_1,\ldots,i_k)} \uparrow^{(1,\ldots,n)}(a) = \left( \sum_{i=1}^{N_{\mathcal{T}}} \sum_{j=1}^{M_{\mathcal{T}}} f_j \uparrow^{(1,\ldots,n)}(a) \right) \left( \sum_{i=1}^{N_{\mathcal{T}}} \sum_{j=1}^{M_{\mathcal{T}}} \mathbb{1}_{A_{\{i\}}}(a) \mathbb{1}_{B_j}(a) \right) \tag{61}$$

Consequently, promotion corresponds to multiplying the embedded activation with indicator functions on the sets of permutations that preserve the reference domains.

### 6.1.6 Writing a k'th order network as an Autobahn

To prove that this bound is tight, we construct an Autobahn network that can emulate the operations in Subsubsection 6.1.1. To do this, we take as our templates the set of all edgeless graphs of size less than or equal to $k$. Consequently, each neuron corresponds to one (ordered) set of $k$ or less nodes. We will then associate each set of neurons to a homogeneous space of the corresponding size. As the automorphism group of these is simply the permutation group applied to the corresponding nodes, the network is able to trivially perform the convolution in (25). To recover (26) we apply narrowing to move the activation from all larger neurons of size $k$ to a smaller neuron of size $m$, $\mathfrak{n}_l$. We simplify this by only considering larger neurons whose reference domain strictly includes $\mathfrak{R}_l$: while this is not necessary, it simplifies our construction. In this case, it is easy to see that the narrowed activation corresponds to the sum over all possible strings of the remaining $k - m$ indices, which corresponds to (26).

Finally, promotion can be used to move an activation from a smaller homogeneous space to a larger one. This follows from (61). Since there is one neuron corresponding to each set of $m$-node graphs, there is exactly one $f_l$ that is nonzero for any given of $a$. Consequently, the value of the function for any given value is precisely the value of $f_l \uparrow^{(1,\ldots,n)}$.

## 6.2 Comparison with GSN

Both the GSN architecture and Autobahn directly featurize graphs using the isomorphism classes of subgraphs. However, whereas GSN uses the isomorphism classes to featurize an MPNN architecture, in Autobahn the neurons convolve directly over the corresponding automorphism group. Consequently, it is natural to ask whether the automorphism group adds any additional flexibility compared to merely using isomorphism to construct features. The following result answers in the affirmative.

**Theorem 4.** *Consider a GSN network with a given set of subgraphs chosen to construct initial features. Next, consider an analogous Autobahn network where each layer's neurons consist of (1) neurons that operate exactly the same way as the MPNN neurons in the GSN (2) Autobahn neurons that on the subgraphs used to featurize the GSN. The Autobahn network is at least as expressive as the GSN network. Moreover, there exists collections of GSN subgraphs such that the Autobahn network is strictly more expressive.*

To prove that the Autobahn network is as expressive as the GSN, we show that it can imitate the behavior of the GSN. Since it has the same MPNN neurons as the GSN, it is enough to show the neurons operating on the subgraphs are able to learn a unique hash for each subgraph. However, this is trivially true since each subgraph has its own weights. To show that the there exist subgraphs such that the Autobahn network is strictly more expressive, it is enough to give a single example. Consequently, we again consider templates corresponding to the edgeless graph of size $k$ or less. Since each node in an $N$-node graph is in exactly $N$ choose $k$ subgraphs isomorphic to the edgeless graph. Each node receives exactly the same feature from the featurization graphs, so this does not improve GSN expressiveness over 1-WL. However, by the results above, the resulting Autobahn can recover a a $k$'th order network by simply ignoring the MPNN neurons. Using results from [7] the network is consequently as powerful as the $k$-WL test, and is consequently strictly more powerful than the corresponding GSN. Note that we cannot say that the Autobahn network is always strictly more expressive than the GSN network because of pathological counterexamples, e.g. setting all of the GSN's subgraphs to be the subgraph of a single node. In this case, the two networks trivially have the same expressivity.

### 6.2.1 Connection with the Reconstruction Conjecture

Previous work has analyzed the flexibility of neural networks in the light of the reconstruction conjecture[4, 8]. In the reconstruction conjecture, one takes a graph $\mathcal{G}$ and forms the multiset of all subgraphs created by deleting a single node from $\mathcal{G}$. The conjecture then states that if two graphs have the same multiset then they must be isomorphic. In [1], it was noted that if the reconstruction conjecture held, a GSN network featurized on all subgraphs of size $N - 1$ to be able to distinguish to isomorphic graphs. However, the performance of the network if the reconstruction conjecture does not hold is unclear.

In contrast, Autobahn is able to reconstruct a graph from neurons operating on each of the $N - 1$-node subgraphs even if the reconstruction conjecture does not hold. We demonstrate this by outlining an explicit algorithm that achieves the reconstruction.

1. The output of the first layer is a single edge feature that is 1 for all edges in the subgraph.

2. Apply a single round of narrowing and promotion. Since narrowing averages over all nodes that are outside of the intersection of the two sets, any edges that are outside the destination local graph are averaged to node features.

3. Pick an arbitrary neuron. Since only edges to nodes outside of the neuron's local graph were truncated, and there is exactly one node missing from the subgraph, connecting the missing node to any nodes that have nonzero node features reconstructs the graph.

Note that this construction does not require any convolutions over the Automorphism group. Instead, the Autobahn network is able to perform the reconstruction due to its ability to perform higher-order message passing.

## 7 Architecture, hyper-parameter and computational details

We provide some further details into the architecture, choice of hyper-parameters and training regime of our network.

### 7.1 Architecture details

We model our block specific (i.e. cycle / path) convolutions after standard residual convolutional networks [3] with ReLU activations. We refer the reader to our pytorch implementation for details of the implementation.

| Modification | Validation loss |
|---|---|
| Original | $0.124 \pm 0.001$ |
| No cycles | $0.175 \pm 0.004$ |
| Only maximum length paths | $0.167 \pm 0.001$ |

Table 1: Validation loss for modified architectures on the Zinc (subset) dataset.

| Dataset | Channels | Dropout | Epochs | Warmup | Decay milestones |
|---|---|---|---|---|---|
| Zinc (subset) | 128 | 0.0 | 600 | 15 | 150, 300 |
| Zinc | 128 | 0.0 | 150 | 5 | 40, 80 |
| MolPCBA | 128 | 0.0 | 50 | 5 | 35 |
| MolHIV | 128 | 0.5 | 60 | 15 | N/A |
| MolMUV | 64 | 0.0 | 30 | 5 | N/A |

Table 2: Hyper-parameter and training schedules used on each dataset.

We perform a simple ablation study of the main components of the model on the Zinc-subset problem. In particular, we study the impact of omitting: i) the cycle-based convolutions and ii) the paths shorter than the maximum length considered. Validation loss[2] results are reported in table 1.

## 7.2   Hyper-parameter details

Our hyper-parameters were chosen based on a combination of chemical intuition and cursory data-based selection, with some consideration towards computational cost. As our architecture specification is quite general, the number of hyper-parameters is potentially large. In practice, we have restricted ourselves to tuning three parameters: a global network width parameter (which controls the number of channels in all convolutional and linear layers), a dropout parameter (which controls whether dropout is used and the amount of dropout), and the training schedule. The values used are specified in table 2. The training schedule is set with a base learning rate of 0.0003 at batch size 128 (and scaled linearly with batch size). The learning rate is increased linearly from zero to the base rate during the specified number of warmup epochs, and is then piecewise-constant, with the value decaying by a factor of 10 after each milestone.

The lengths of the paths and cycles considered in the model are also hyper-parameters of the model. We used the same values (cycles of lengths 5 and 6, and paths of lengths 3 to 6 inclusive) in all of our models. We note that in molecular graphs, cycles of lengths other than 5 and 6 are exceedingly rare (e.g. in the Zinc dataset, cycles of lengths different from 5 or 6 appear in about 1% of the molecules). We evaluate different possibilities for the maximum length of paths to be considered in table 3, we observe that in general, both computational time and prediction performance increase with larger path lengths.

## 7.3   Computational details

In a message passing graph neural network, computational time is typically proportional to the number of edges present in the graph. On the other hand, our Autobahn network scales with the number of paths and cycles present in the graph. An immediate concern may be that the number of such structures could be combinatorially large in the size of the graph. In table 4, we show that, due to their tree-like structure, molecular graphs do not display such combinatorial explosion of number of sub-structures in practice.

The computational cost of our model scales roughly linearly with the total number of substructures under consideration. In practice, for molecular graphs, selecting only paths of short lengths and cycles, we expect

---

[2]Note that for this particular dataset, validation losses tend to be higher than test losses, which is also observed in some other architectures such as HIMP.

| Path length | Validation Loss | Training Time |
|---|---|---|
| 4 | $0.140 \pm 0.000$ | 3 h 15 min |
| 5 | $0.135 \pm 0.000$ | 4 h 50 min |
| 6 | $0.124 \pm 0.000$ | 6 h 50 min |
| 7 | $0.121 \pm 0.001$ | 9 h 30 min |

Table 3: Performance and training time of model on Zinc (subset) as a function of maximum path length considered.

the computational cost to be on the same order of magnitude as standard graph neural networks. We report the total amount of time (in GPU-hours) required for training each of our model in table 5. The training was performed on Nvidia V100 GPUs, and mixed-precision computation was used for all models except MolHIV were some gradient stability issue were encountered. The two largest datasets (Zinc an MolPCBA) were trained on four GPUs, whereas the remaining datasets were trained on a single GPU.

| Dataset | Nodes | Edges | Paths | | | | | | Cycles | |
|---|---|---|---|---|---|---|---|---|---|---|
| | | | 3 | 4 | 5 | 6 | 7 | 8 | 5 | 6 |
| Zinc | 23.1 | 49.8 | 34.6 | 43.9 | 55.0 | 64.4 | 65.8 | 70.2 | 0.56 | 1.70 |
| MolPCBA | 26.0 | 56.3 | 39.3 | 51.0 | 65.2 | 79.5 | 84.2 | 93.1 | 0.50 | 2.23 |
| MolHIV | 25.5 | 54.9 | 39.2 | 52.1 | 68.9 | 87.2 | 97.2 | 111.5 | 0.34 | 2.01 |
| MolMUV | 24.2 | 52.5 | 36.5 | 47.6 | 61.1 | 73.4 | 77.2 | 84.6 | 0.63 | 2.02 |

Table 4: Average count of structures in various datasets.

| Dataset | Samples | Samples / s / GPU | Training time (GPU-hours) |
|---|---|---|---|
| Zinc (subset) | 6.0M | 280 | 6.8 h |
| Zinc | 33.0M | 215 | 42.7 h |
| MolPCBA | 21.9M | 171 | 35.5 h |
| MolHIV | 2.5M | 114 | 6.1 h |
| MolMUV | 2.8M | 222 | 3.5 h |

Table 5: Computational cost of training provided models. Samples denotes total number of gradients computed (i.e. number of epochs times number of observations in dataset).