# OpenReview forum: "Autobahn: Automorphism-based Graph Neural Nets"
_NeurIPS.cc/2021/Conference — NeurIPS 2021 Poster_

### Official Review · Reviewer_NsaB · 2021-07-16

**Rating:** 5
**Confidence:** 2

**Summary:**

This paper proposed Autobahn, a new GNN model for learning representations of graphs for graph classification tasks. The idea is to extract subgraphs that are isomorphic to predefined graph templates (e.g., cycles, paths) and perform a convolution operation on its automorphism group. This paper proposed two operations to perform interaction between the representations on different subgraphs: narrowing and promoting. This paper applied the proposed model to graph classification tasks on a molecular graph dataset and verified its effectiveness.

**Ethical Concerns:**

N.A.

**Limitations And Societal Impact:**

- [L1] This paper discussed the technical limitations (l.450)
- [L2] This paper discussed possible negative social impact in the broader impact section (l.334)

**Main Review:**

### Soundness (Do theorems and experiments answer research questions, assuming they are correct?)

- [S1] The research question of this paper is to construct GNNs that exploit the topological information of graphs and to use it to learn effective representations of molecular graphs.
- [S2] From a theoretical point of view, Autobahn is an answer to the research question. It is a model that explicitly utilizes structural information in its construction.
- [S3] On the other hand, although Autobahn is an extension of MPNN, if my understanding is correct, there is no theoretical support for whether Autobahn learns molecular graph representations better than MPNN. For example, it is not known that Autobahn is strictly more expressive than MPNN.
- [S4] Regarding empirical evaluation, the ZINC dataset and the Open Graph Benchmark are typical benchmarks for molecular graph tasks. Therefore, I think the experiments are good evidence that Autobahn empirically extracts molecular graph representations.

### Correctness (Are derivation of theorems and experiments correct?)

- [C1] As far as I checked, I did not find significant mistakes in the proofs or experimental setup. However, I have several questions regarding the mathematical description of the proposed model. Here are some examples.
- [C2] In Figure 2, the method matches the input graph to the template graph $T$ using $\mu=(1, 3)$. However, I think the choice of $\mu$ is not unique. In the case of Figure 2, we can also use $\mu=(2, 1, 6, 3, 4, 5)$ (or its inverse?) to match the graphs.
- [C3] l.168: It is not clear to me how we define the local graph $\mathcal{G}^\ell_j$ for the node $j$. At first sight, I thought that the local graph for the ndoe j is a subgraph containing the node j such that it is isomorphic to some template graph. However, I am not sure how we should define $\mathcal{G}^\ell_j$ when we find two subgraphs that match the template(s).
- [C4] I understood that Algorithm 1 is the aggregation of an MPNN when the template graph is a star graph (Section 5.1). However, Autobahn has narrowing and promotion operations, too. Therefore, it is unclear whether Autobahn includes MPNN as a special case (the same is true for Steerable CNNs).

### Novelty and Significance (Do the paper have novel points? If so, are they significant?)

- [N1] The idea of utilizing subgraph information into graph representation learning appeared in several existing studies such as Graph Homomorphism Network (GHN) [Nguyen and Maehara, ICML2020], Graph Substructure Networks (GSN) [Bouritsas et al. al., arXiv], and Structural Message Passing (SMP) [Vignac et al., NeurIPS2020]. In particular, GSN and SMP are similar to this paper in that they are extensions of MPNN.
- [N2] However, I think this paper has novel points compared with these models. Specifically, existing models incorporated subgraph information explicitly (GHN, GSN) or implicitly (SMP) for creating input features (i.e., feature extraction stage). On the other hand, this paper tried to incorporate subgraph structure into convolutional operations.
- [N3] If I understand correctly, Autobahn performed an equivariant message passing on two isomorphic local subgraphs in an input graph. Natural Graph Neural Network also employed the idea of equivariant message passing [de Haan et al., NeurIPS2020]. I would like the authors to clarify and discuss the relationship between Autobahn and Natural Graph Neural Network.
- [Nguyen & Maehara, ICML2020] http://proceedings.mlr.press/v119/nguyen20c.html
- [Vignac et al., NeurIPS2020] https://proceedings.neurips.cc//paper_files/paper/2020/hash/a32d7eeaae19821fd9ce317f3ce952a7-Abstract.html
- [Bouritsas et al., arXiv] https://arxiv.org/abs/2006.09252
- [de Haan et al., NeurIPS2020] https://proceedings.neurips.cc//paper/2020/hash/2517756c5a9be6ac007fe9bb7fb92611-Abstract.html

### Clarity (Is the paper clearly written?)

- [C1] The organization of the paper is OK. The structure of this paper is in line with the reader's understanding, starting with a simple case (Section 4) and then explaining the general case (Section 5).
- [C2] However, it took me much effort to figure out what the description of the proposed model mathematically means. In addition, there were many cases in which I could not understand the text just by reading it once.
- [C3] In addition, I could grasp the intention of some part of the text after completing reading the whole paper and understanding the architecture of Autobahn (e.g., l.41).

### Other comments

- l.87: One way to accomplish this would to be ... → would be ...
- l.90: $\phi_l$ → $\phi_{\ell}$
- l.107, Eq. (6): If I understand correctly, Eq. (6) seems to be a bit of an abuse of notation. Specifically, in the left-hand side, $(f\ast w)_j$ is the value at the point $j$ in the underlying space. However, in the right-hand side, $f$ and $w$ are interpreted as functions on the group which acts on the underlying space.
- l. 187, Eq. (9): There seems some typos. First, $w_\sigma(1)$ in the first term on the right-hand side has only one subscript, while $w_{jk}$ in the second terms has two subscripts. Also, I think $\sum_j$ is supposed to be $\sum_k$.
- l.196: I understand that MPNNs have a central node. But I am not sure what the central node means in CNNs.
- l.257: \tilde{f}\_z → \tilde{f}\_{s\_z} (two places)
- Table 1: compared with ... → Compared with ...

### Post-rebuttal comments

The authors answered my questions and they solved most of my questions. In addition, I admit the novelty and significance of this paper. However, I still think that this paper is hard to access for those who are not familiar with this topic, although I understand that the authors have made considerable efforts to improve the readability. Therefore, I want to keep my score (5) and ask the authors to enhance the accessibility.

**Time Spent Reviewing:**

12

---

> ### Author Response · Authors · 2021-08-10
> **Response to Reviewer NsaB**
>
> We thank the reviewer for the time and effort they have put into this review.  They have asked several interesting questions, which we have responded to below in the order they were presented.
>
> **Soundness**
>
> [S3] To address the reviewer’s comment on the relative expressiveness of Autobahn and MPNNs, it is not true that all Autobahn networks are more expressive than MPNNs. As MPNNs are a subset of Autobahns, this would lead to a contradiction.  However, there do exist Autobahn networks that are more expressive than MPNNs.  As other reviewers have also asked us about theoretical results on the expressivity of Autobahn architectures, we have given some simple theoretical results in a separate comment.
>
> **Correctness**
>
> [C2] The choice of the matching permutation $\mu$ is indeed not unique: by definition, it is only defined up to an element of the graph’s automorphism group.  It is for precisely this reason that the automorphism group is so important to our formalism.  If our convolutions are equivariant to the graph’s automorphism group then using different matchings in Algorithm 1 gives the same result.
>
> [C3] In principle, any selection procedure is acceptable as long as the three properties described in Theorem 1 hold. In practice, a simple choice is to simply list every subgraph that is isomorphic to a template, and let the index $j$ denote the $j$’th subgraph in the resulting list.  One could certainly imagine cases where the ordering of the subgraphs and the nodes coincide (as they do in MPNNs) but this is not a general requirement. We will add a few sentences in the revision near line 275 to ensure that this is clear to the reader.
>
> [C4] Rather than thinking of an MPNN as having nodes that send messages to each other, we will describe MPNNs as a composition of two operations occurring on all of a graph’s star-shaped subgraphs.  The first operation applies a computation that aggregates features from a star’s leaves and puts the result on the star’s center.  This is a specific example of an automorphism-equivariant convolution.  The second operation then copies this node feature to every other overlapping star.  This is an example of narrowing and promotion.  It is easy to see that for two different overlapping star graphs, this copying procedure only transmits information if the centers are connected by an edge.  Consequently, the old center features become the new leaf features for the next iteration.  Composing these two steps will aggregate information over edges and then pass it to neighboring nodes in subsequent steps, performing the same steps as in an MPNN.  The procedure for writing steerable-CNNs as an Autobahn architecture is analogous, but slightly more complicated.  We refer the reviewer to the corresponding section of the supplement.
>
> **Novelty and Significance**
>
> As the reviewer has pointed out, several previous networks have used the local graph structure to featurize existing GNN paradigms such as MPNNs. Moreover, there has also been considerable work on MPNN-like architectures with components that are equivariant to local permutations, e.g. in References [1,3,5,6].  Our work builds on this research by constructing a formalism (Autobahn) where subgraphs are the fundamental unit of computation.  This represents a move away from the node and edge-based approach common to most GNN architectures.
>
> One key component in our approach is the observation that it is often enough to preserve equivariance to a graph’s automorphism group rather than equivariance to the set of all permutations.  This realization is originally due to de Haan et al. [1], and we attribute it to them in Section 4.  However, the two works then take these realizations in different directions.
>
> In de Haan et al., the authors propose a framework known as Local Natural Graph Networks (LNGNs).  The LNGN framework follows the central idea in MPNN architectures of having node features that communicate over edges.  LNGN builds on this by associating each node and edge with a subgraph called the node or edge neighborhood which corresponds to its local environment.  Communication between nodes then happens in an environment aware-manner.  However, while the authors give automorphism-equivariance as a minimal constraint on the computation, all of the GNN architectures proposed in de Haan et al. actually use message-passing neural networks restricted to local neighborhoods.   As a result, the subgraph computations used in LNGNs actually obey the much stronger constraint of local _permutation_ equivariance.
>
> In fact, at the beginning of Section 4 the authors actively advocate against the using Autobahn-like neurons in LNGNs, as they believe the neighborhoods constructed by LNGNs are too heterogeneous for such a strategy to be practical.  We agree, and believe this heterogeneity is a direct consequence of the node and edge-based approach that forms the basis of the LNGN framework:  if we view LNGNs through lens of the Autobahn formalism, it becomes clear that the identification of subgraphs with nodes and edges imposes strong combinatorial constraints on which subgraphs can be used without breaking global permutation equivariance.
>
> Autobahn differs from LNGNs (and indeed most GNNS) by making subgraphs the fundamental unit of computation as opposed to individual nodes or edges.  The shift from a node/edge based picture to explicit computation on generic subgraphs is one of the key theoretical novelties in our work.  (Other theoretical novelties include the definition of the narrowing and promotion operators as well as a formal explanation of how MPNNs operate from the perspective of group-equivariant neural networks.)  Since Autobahn works directly with the subgraphs themselves, Autobahn networks can work with more general families of subgraphs than LNGNs.  In particular, we can choose families of subgraphs that are much more homogeneous than local node neighborhoods.   Consequently, we are able to construct practical GNN architectures where neurons use explicit convolutions against the automorphism group.  To our knowledge, we are the first to do so.
>
> Ultimately we believe the statement given in our paper (that the GNN architecture ultimately used in de Haan et al. does not explicitly leverage the automorphism structure of a graph or its parts) is a succinct summary of the most salient difference between the two works.  However, we would be happy to expand further on the differences between the LNGNs and Autobahn in the main text.
>
> **Other Comments**
>
> The miscellaneous comments noted by the reviewer were good catches.  Three in particular deserve special response.
> - Equation (6) is indeed an abuse of notation: we had sneakily exploited a bijective mapping between rotations and graph nodes.  We will correct this in revision.
> - The reviewer is right to be confused by equation (9): we made a math mistake when writing it and the equation is incorrect.  While fixing (9) is an option, we think it would be even better to simply refer the reader to Equivariant MLP library [2] rather than asking them to write the code themselves.
> - By central node in a CNN we had meant the central pixel when a given template is applied (see Section 3 of the supplement).  We will reword this sentence to make this clearer.
>
> **References**
>
> [1] de Haan, Pim, Taco S. Cohen, Max Welling. “Natural Graph Networks” Advances in neural information processing systems (2020).
>
> [2] Finzi, Marc, Max Welling, and Andrew Gordon Wilson. "A Practical Method for Constructing Equivariant Multilayer Perceptrons for Arbitrary Matrix Groups." arXiv preprint arXiv:2104.09459 (2021).
>
> [3] Kondor, Risi, et al. "Covariant compositional networks for learning graphs." arXiv preprint arXisbv:1801.02144 (2018).
>
> [4] Maron, Haggai, et al. "Invariant and Equivariant Graph Networks." International Conference on Learning Representations. 2018.
>
> [5] Vignac, Clement, Andreas Loukas, and Pascal Frossard. "Building powerful and equivariant graph neural networks with structural message-passing." arXiv preprint arXiv:2006.15107 (2020).
>
> [6] Zhengdao, Chen, et al. "Can Graph Neural Networks Count Substructures?." Advances in neural information processing systems (2020).

---

> > ### Comment · Reviewer_NsaB · 2021-08-29
> > **Thank you for your response.**
> >
> > I thank the authors for the detailed responses.
> >
> > [C2]: OK
> >
> > [C3]:
> > > In practice, a simple choice is to simply list every subgraph that is isomorphic to a template, and let the index j denote the j’th subgraph in the resulting list.
> >
> > Does it mean that each node can be assigned to more than one subgraph?
> >
> >
> > [C4]: Although it is a minor point, I have a question about this part of the author's reply
> > > The first operation applies a computation that aggregates features from a star’s leaves and puts the result on the star’s center. This is a specific example of an automorphism-equivariant convolution
> >
> > I think features between leaf nodes and the center node do not interact by the group convolution for a star graph. Suppose space X is divided into multiple orbits by the action of G. According to Appendix 1, the group convolution operates on each orbit and concatenates the results on the orbits. The star graph is divided into two orbits (leaf nodes and the center node). Therefore, the leaf information is not transferred to the central node by the group convolution alone. I think we need narrowing and promoting to mix representations of leaf nodes and the center node. Let me know if my understanding is not appropriate.
> > I would like to suggest explaining how Autobarn is reduced to MPNN in more detail in the case of star graphs.
> >
> > [Novelty and Significance]: I understand that Section 4 summarizes the differences between Autobahn and [de Haan et al., 20]. I appreciate the elaboration of the discussion about the differences.

---

> > > ### Author Response · Authors · 2021-08-30
> > > **Further discussion of the details of Autobahn**
> > >
> > > > I thank the authors for the detailed responses.
> > >
> > > Our pleasure! We're quite passionate about this line of research and happy to have an opportunity to discuss it more.
> > >
> > > [C3]:
> > >
> > > > Does it mean that each node can be assigned to more than one subgraph?
> > >
> > > Indeed: each node can be in arbitrarily many subgraphs. For instance, in Figure~3 nodes 1 and 4 are in both the local graphs associated with neuron $i$ and neuron $j$. Similarly, in Figure 4 the top-most and top-right atoms in the aromatic ring (i.e. the node with the darkest green activation and its neighbor to the bottom right) are depicted in two of their subgraphs: a cycle subgraph and a path subgraph.
> > >
> > > [C4]:
> > >
> > > > I think features between leaf nodes and the center node do not interact by the group convolution for a star graph. Suppose space X is divided into multiple orbits by the action of G. According to Appendix 1, the group convolution operates on each orbit and concatenates the results on the orbits. The star graph is divided into two orbits (leaf nodes and the center node). Therefore, the leaf information is not transferred to the central node by the group convolution alone. I think we need narrowing and promoting to mix representations of leaf nodes and the center node. Let me know if my understanding is not appropriate. I would like to suggest explaining how Autobahn is reduced to MPNN in more detail in the case of star graphs.
> > >
> > > We think we understand the source of the confusion: group-convolution is able to transfer information between orbits by taking linear combinations over corresponding values. (Indeed, this is exactly what channel-mixing corresponds to from a group-theoretic perspective). It might be helpful to think of the example discussed at the bottom of page 4, where we consider a graph whose automorphism group is the trivial group. Each node is in its own orbit, and its node feature corresponds to the same group element (the identity element) when viewed as a function on the group.  Since we can freely mix features corresponding to the same group element, we can just run a fully-connected network on each of the node features.
> > >
> > > Let us know if this helps clear things up. Relatedly, we’d be happy to add a further section to the supplement that breaks down in detail how to construct an Autobahn network that mimics an MPNN. Diving into this topic in depth might help interested readers get a firmer grip on our core ideas, and could also be educational for scholars who aren't as familiar with the minutae of group-equivariant networks.

---

> > > > ### Comment · Reviewer_NsaB · 2021-08-31
> > > > **Thank you.**
> > > >
> > > > Thank you for your detailed explanation. I think I can clarify my question.
> > > >
> > > > [C3] OK
> > > >
> > > > [C4] OK. I think I understand that we can transfer information between the center and leaf nodes by concatenating features on the same group element and taking their sum by channel-wise mixing.

---

> > > > > ### Author Response · Authors · 2021-09-01
> > > > > **Response to Reviewer NsaB**
> > > > >
> > > > > We are glad we were able to clarify these questions, and are happy to answer any other questions you might have as you continue to re-evaluate our work.

---

> ### Author Response · Authors · 2021-09-03
> **Response to Post-rebuttal Comments**
>
> We appreciate that the reviewer agrees with us about the novelty and significance of this work.
>
> However, we disagree with the criticism that our paper should be rejected because it is hard to access for those who are not familiar with this topic.  Like any research article, we draw from a well-established body of literature, and familiarity with that literature is a prerequisite to understanding our paper.

---

### Official Review · Reviewer_jtWY · 2021-07-16

**Rating:** 7
**Confidence:** 4

**Summary:**

This paper proposes Automorphism-based GNNs (Autobahns), a framework that generalises GNNs from operating in star-like neighbourhoods to arbitrary local subgraphs.  In particular, the authors use group-theoretic tools to define linear operators on arbitrary graphs and show that their parameters are constrained by the automorphism group of the graph. However, as also discussed in Natural Graph Networks (de Haan et al., NeurIPS’20), defining a different operator for each graph would make learning impossible since there would be virtually no weight-sharing across training data (generalisation to unseen graphs would be also impossible). To this end, the authors deal with a set of predefined small-sized template graphs on which they define their learnable operators. Then each graph is processed as follows: first, it is decomposed into a collection of overlapping templates. Then, (1) the components (e.g., nodes and edge features) of each matched subgraph are transformed by the operator of the corresponding template and (2) overlapping subgraphs “exchange messages” via the narrowing and promotion operations, as defined by the authors.  Steps (1) and (2) are repeated for each layer. The proposed framework is instantiated with cycle and path templates and evaluated on molecular datasets achieving competitive results.

**Limitations And Societal Impact:**

*Limitations*: The authors do mention the limitation of the template graph selection and make a brief reference on the subgraph isomorphism problem they have to solve. As I mentioned in my main review, the latter, together with the training complexity, should be more clearly stated.

*Negative social impacts*: Addressed in Sec. 7.1.

**Main Review:**

This paper falls under the category of structure-aware GNNs. In particular, ever since the first impossibility results were proposed for traditional message passing, most notably the fact that its expressivity is upper bounded by the 1-Weisfeiler Leman test and that it cannot count the vast majority of substructures, several extensions have been proposed in order to increase the expressivity of GNNs by providing them with richer structural information (see below for details). This paper takes a similar approach making use of the group-theoretic perspective of convolutions as formalised in Cohen and Welling, “Group Equivariant Convolutional Networks”, ICML’16 and Kondor and Trivedi, “On the generalization of equivariance and convolution in neural networks to the action of compact groups”, ICML’18.

**On the positive side**:

-	The proposed framework is general enough and (although not proven by the authors) clearly more expressive than conventional message passing.
-	Moreover, group theoretic formulations are elegant and provide a principled way to define neural operators in the presence of symmetries. Section 4 is quite intuitive and provides insights in how to define expressive linear operators over simple structures that are governed by symmetries.
-	The accompanying illustrations (Figure 2, 3, 4) improve the readability and the accessibility of this work to a larger audience.
-	The cycle/path-Autobahn instantiation seems to work relatively well in real-world molecular data.

However, I have some reservations because (1) the proposed method has certain drawbacks that may limit its applicability and (2) some of the ideas in this paper are not entirely new and unfortunately the authors have not adequately discussed the connections of their work with the relevant literature. I believe it would be beneficial (and perhaps necessary) to draw these connections at a conceptual level and provide some theoretical evidence to support the merits of the approach. In particular:

 **Applicability concerns**:

1.  The method cannot be seamlessly extended to arbitrary graph templates, since one needs to derive a new automorphism-equivariant operator every time a new template is added,
2.  it can become computationally cumbersome during training (for subgraphs of up to k nodes one may encounter ${n \choose k}$ subgraphs, where $n$ the number of nodes).
3.  It requires selecting the subgraphs by domain knowledge.

**Related Work**:

1. Most notably the paper by de Haan et al., “Natural Graph Networks”, NeurIPS'20 proposed quite similar ideas with Autobahn with small differences (e.g., in NGN the authors take into account all edge-wise isomorphism classes, which makes it quite hard to generalise, and computation is performed at node/edge level, while here it is performed at a subgraph level). I think the authors should dedicate a few sentences to clearly compare with this paper.
2. Additionally, in some papers, subgraph information is incorporated in the message passing in the form of subgraph counts (e.g., GSNs: Bouritsas et al., “Improving graph neural network expressivity via subgraph isomorphism counting”, arxiv’20). The authors refer to this method in L78, but in my opinion, the comparison is insufficient and potentially interesting connections remain unidentified. GSNs also share the same limitation w.r.t. the selection of the template subgraphs, but they are more computationally efficient (the same computational burden is encountered for the pre-processing in both methods, but during training, GSN is $O(m)$, where $m$ the number of edges), it is easier to implement and can be seamlessly extended to arbitrary subgraphs. I suspect that Autobahns can be proven to be more expressive than GSNs, since each node does not only possess the information of the number of subgraphs it belongs to, but also its position in each subgraph. This brings me to my next concern.

**Theoretical evidence** :

In my opinion, the paper would benefit a lot from a theoretical analysis w.r.t. the expressive power of Autobahns. Currently, the method is reasonable at an intuitive level, but I suspect that strong expressivity results can be obtained (which would outweigh the above drawbacks), and thus I would encourage the authors to look into this direction.

**Experiments**:
1.  More baselines should be added (see the leaderboards for ZINC and ogb),
2. The methods should be sorted according to their performance,
3. In ZINC, what is the number of parameters (according to the protocol the number of parameters should be ~100K or ~500K)?
4. Perhaps it would be better to relax a bit the SOTA claims (Autobahn is on SOTA ZINC – on par with GSN, but not in the other datasets). Just to clarify, this is not an issue (at least for me), but right now the claim is misleading.

**Additional comments**:

-	Section 5 is hard to follow and the method seems to be quite involved, although I am not sure if this is actually the case, especially for the template graphs the authors use in practice. In particular,
   - Eq. (9) should have been explained more thoroughly and the notation is a bit confusing. Currently, it is not very intuitive. Should the index of the summation in the second term have been $k$? Maybe also $f_{\sigma(k)}$ --> $f_{\sigma^{-1}(k)}$.
  -  A practical example (e.g., for a cycle graph) would help.
  - The notation and the explanations of the narrowing and promotion operations are, at least to me, very hard to follow.
  - L229: the notation is confusing
-	If I understand correctly from Algorithm 1, this approach requires matching the nodes of the subgraph to the template, which is NP-complete, unless the size of the template is constant. I can see that the authors mention that in a footnote in P5 and thus it might go unnoticed. I think it should be clearly stated along with the worst-case training complexity.
-	What happens if a node does not belong in any of the subgraphs?
-	I am confused with the claim of L116, but maybe I misunderstood. In case one deals with a single graph, using the general group convolution of Eq. (5) one should obtain different weights per vertex (as in the global version of natural graph networks and in section 4 of this paper). Is that correct?


**Justification of rating**: To summarise, I like the approach, I find it elegant and it provides a generic recipe for convolutions in arbitrary graphical structures. However, the limitations and drawbacks of the approach should be clearly stated, along with a more elaborate comparison of the related work. Moreover, in my opinion, given that the experiments do not show a clear advantage of the approach, I think that theoretical evidence will improve the impact of the paper a lot. At this stage, I am not sure if the paper is ready for publication. Hence, I will choose a baseline rating that I am happy to change after the discussion with the authors.


------
## After rebuttal

The authors did provide theoretical evidence that supports and explains their approach, hence I have increased my score and I will recommend acceptance. I still have some reservations given that this evidence is a major modification to the paper and because of the fact that the method still suffers from some limitations w.r.t. its applicability. Please see my comment below for more information



**Time Spent Reviewing:**

12

---

> ### Author Response · Authors · 2021-08-10
> **Response to Reviewer jtWY**
>
> We thank the reviewer for their detailed investigation of our paper.  In particular, we appreciate the reviewer’s opinion that our framework is an elegant and principled approach to building graph neural networks (GNNs)
>
> The reviewer’s comments fall into four main areas: applicability, connection to related work, theoretical evidence, and experimental results.  Regarding the experimental results, we are happy to make the minor changes suggested.  The other areas require further discussion; we will address them below.
>
> **Applicability**
>
> -  While the task of deriving new automorphism-equivariant operators might seem daunting, recently developed code libraries make this task much less onerous.  For instance, the equivariant-MLP library [4] automatically constructs convolutions against arbitrary groups, and the igraph library [5] automatically finds the generators of a graph’s automorphism group.  Moreover, for most graphs the automorphism group is actually quite simple, often even the trivial group.
> - While it is true that Autobahn operates on $\binom{N}{k}$ subgraphs in the worst case, this combinatorial scaling is a feature of most (if not all) graph neural networks that use multi-node hidden representations, e.g. [6,7,8,10].   As such, we think this is not a unique feature to Autobahn as much as a generic theme of this research direction.  Moreover, we note that if Autobahn templates are connected graphs, then there are at most $n * d^{(k-1)}$ templates where $d$ is the graph’s maximum degree.  In any case, we agree that it would be good to discuss this explicitly and we will include a brief discussion in the revised paper.
> - Rather than saying that Autobahn requires selecting subgraphs through domain knowledge, we think it is more accurate to say that Autobahn gives users the option to use domain knowledge to select better subgraphs.  As discussed in the text, there are generic subgraph choices (e.g. star graphs and paths) that require little-to-no domain knowledge.  However, when domain knowledge is available, it seems natural to use it.  We believe this idea is implicit in most successful machine learning approaches.  For instance, we do not use MPNNs on image pixels but instead explicitly leverage the grid structure of the image.  Consequently, we think the fact that Autobahn lets users use very structured templates is not a weakness but a strength.
>
> **Connection to related work**
>
> *Connection with NGNs*
>
> de Haan et al. is a very natural point of comparison with our work.  With the added space available here, we are happy to discuss the difference between these topics in depth.
>
> To our knowledge, de Haan et al. [3] is the first paper to observe that leveraging a graph’s automorphism group can potentially lead to more expressive architectures, something we credit them for in our work.  However, apart from this observation, LNGNs [3] and Autobahn approach graphs in different ways.  LNGNs have the same philosophy as MPNNs: they treat the graph as a collection of nodes that interact through edges.  Indeed, we suspect that one result of this approach might be an architectural bias towards node and edge features alluded to by the reviewer.
>
> A much more important consequence is that whenever subgraphs are used in LNGNs, they are constructed as edge and node neighborhoods.  Even for the relaxed definition of neighborhood used in Appendix C, the injective map from nodes/edges to subgraphs imposes a strong combinatorial constraint on which subgraphs can be used without breaking global permutation equivariance.  Specifically, for a graph of $N$ nodes and $M$ edges, LNGNs have at most $N$ node neighborhoods and $M$ edge neighborhoods.  Moreover, the neighborhoods must be chosen in a manner that is invariant to permutation of the graph.  This constraint is violated for most collections of subgraphs, including the paths and cycles used in our example Autobahn.
>
> As a consequence, LNGNs will generally use a very heterogeneous set of subgraphs: something the authors observe as well. Consequently they do not explicitly use convolutions against the automorphism group. Instead, in LNGNs automorphism-equivariance is treated as a minimal constraint and the action of the neuron is left unspecified. In fact, readers are actually cautioned against using the neurons similar to those used in Autobahn. The GNN architectures the authors propose actually use message-passing neural networks restricted to local neighborhoods.  The architectures given in [3] therefore have neurons that obey a much stronger constraint: local permutation equivariance.
>
> In contrast, Autobahn makes subgraphs the fundamental building block of computation.  The shift from a node/edge based picture to explicit computation on generic subgraphs is one of the key theoretical novelties in our work.  (Other theoretical novelties include the definition of the narrowing and promotion operators and the formal justification of MPNNs as group-equivariant neural networks.)  As a consequence Autobahn does not face the same combinatorial constraints and we can work with a more homogeneous family of graphs.  Consequently, we are able to construct practical GNN architectures where neurons use explicit convolutions against the automorphism group.  To our knowledge, we are the first to do so.
>
> Ultimately we believe the statement given in our paper (that the GNN architecture ultimately used in de Haan et al. does not explicitly leverage the automorphism structure of a graph or its parts) is a succinct summary of the most salient difference between the two works.  However, we would be happy to expand further on the differences between the two formalisms and resulting architectures in the main text.
>
> *Connection with GSNs*
>
> There is indeed a very natural comparison between GSN [1] and Autobahn: in a separate comment, we give some theoretical results directly comparing the two architectures.  It is also interesting to think of combining the two approaches for precisely the efficiency considerations the reviewer mentions. One approach would be to construct GSN-like features on a large class of subgraphs, and then construct Autobahn neurons on a smaller subset of subgraphs.  Alternatively, it might be possible to use Autobahn-like neurons to learn embeddings of labeled subgraphs that could be used as GSN features for subsequent architectures.  In short, we believe the approaches described in the GSN paper and in Autobahn have the potential to complement each other very well.
>
> **Theoretical Evidence**
>
> Initially, we were intending to address the expressivity and relative efficiency of Autobahn architectures in a subsequent paper, giving us more space to comprehensively discuss the topic.  However, the reviewer’s point -- that having some theoretical guarantees in this paper would help motivate and justify our current work -- is well-taken.  To this end, we have given some results that are relatively self-contained in a separate comment.  If the reviewers think these results strengthen the paper we are happy to include them, along with their respective proofs.
>
> **Additional Comments**
>
> - The reviewer is right to be confused by equation (9): we made a math mistake when writing it and the equation is incorrect.  While fixing (9) is an option, we think it would be even better to simply refer the reader to the equivariant-MLP library [4] which accomplishes this task, rather than asking them to write the code themselves.
> - Indeed, our templates are of fixed size and the subgraph matching is polynomial. We are happy to move the footnote into the main text, as well as make additional reference to previous work on algorithms for efficiently extracting isomorphic subgraphs.
> - If nodes do not appear in any of the subgraphs they do not appear in the compute graph.  Depending on the application this may be a failure mode, similar to how MPNNs can “fail” when applied to disconnected graphs.  In any case, this situation is easy to avoid in practice by just adding star or path subgraphs on any “left-over” nodes.
> - We don’t quite understand what is meant by “assigning different weights per vertex.”  Equation (7) is constrained so that the input and output of the convolution are a single list of node features.  In this case there are only two learned parameters possible, as described in [9].  This is because the constraints force $w$ to have the same value for large groups of permutations.  For a further discussion, we refer the reviewer to References [7, 9].
>
> **References**
>
> [1] Bouritsas, Giorgos, et al. "Improving graph neural network expressivity via subgraph isomorphism counting." arXiv preprint arXiv:2006.09252 (2020).
>
> [2] Dong, Yihe, Will Sawin, and Yoshua Bengio. "Hnhn: Hypergraph networks with hyperedge neurons." arXiv preprint arXiv:2006.12278 (2020).
>
> [3] de Haan, Pim, Taco S. Cohen, Max Welling. “Natural Graph Networks” Advances in neural information processing systems (2020).
>
> [4] Finzi, Marc, Max Welling, and Andrew Gordon Wilson. "A Practical Method for Constructing Equivariant Multilayer Perceptrons for Arbitrary Matrix Groups." arXiv preprint arXiv:2104.09459 (2021).
>
> [5] The igraph Core Team. (2021). igraph (0.9.4). Zenodo. https://doi.org/10.5281/zenodo.4884688
>
> [6] Kondor, Risi, et al. "Covariant compositional networks for learning graphs." arXiv preprint arXiv:1801.02144 (2018).
>
> [7] Maron, Haggai, et al. "Invariant and Equivariant Graph Networks." International Conference on Learning Representations. 2018.
>
> [8] Vignac, Clement, Andreas Loukas, and Pascal Frossard. "Building powerful and equivariant graph neural networks with structural message-passing." arXiv preprint arXiv:2006.15107 (2020).
>
> [9] Zaheer, Manzil, et al. "Deep sets." arXiv preprint arXiv:1703.06114 (2017).
>
> [10] Zhengdao, Chen, et al. "Can Graph Neural Networks Count Substructures?." Advances in neural information processing systems (2020).

---

> > ### Comment · Reviewer_jtWY · 2021-08-26
> > **Recommendations after the rebuttal**
> >
> > Following the discussion with the authors, although it is a bit hard to carefully assess the new theoretical evidence, I think a significant effort has been made to provide explanations, hence I have decided to recommend acceptance.
> >
> > However, I would like to stress that in my opinion acceptance should be conditioned on including the theory, or at least a part of it, in the updated version.
> >
> > Final comments:
> > - The authors should highlight the connections of their work with GSN since they are quite similar in spirit.
> > - The authors should be more up-front w.r.t. the practical use of their architecture. My concern about extending to new templates has been addressed, but I am afraid that the architecture still has some important drawbacks:
> >     1. The ${n \choose k}$ complexity will make training slow in many graph distributions such as social networks. The authors correctly say that this problem is not unique to their method, but unfortunately, it has been the main criticism against the methods they cite, and probably the reason why they are not that popular in practice.
> >     2. The computational burden of the preprocessing is in my opinion not adequately discussed. Although I agree that subgraph isomorphism is polynomial in the size of the graph and that in many real-world distributions heuristics provide fast solutions, the authors should acknowledge that their method might be quite slow in some cases such as densely connected graphs.
> >     3. Template selection in my opinion is an issue and should be clearly acknowledged (the authors only experimented with a few molecular distributions, which might have concealed this problem).
> > - Finally, I would like to reiterate my comment that section 5 should be simplified (especially the narrowing and promotion operations).

---

> > > ### Author Response · Authors · 2021-08-26
> > > **Response to recommendations**
> > >
> > > We thank the reviewer for the considerable effort they have put into reviewing this paper: they have shown an admirable dedication to their craft.  We are happy to make the changes suggested by the reviewer and we believe the paper will be stronger as a result.

---

### Official Review · Reviewer_3yjU · 2021-07-16

**Rating:** 6
**Confidence:** 2

**Summary:**

This paper introduces Automorphism-based Graph Neural Networks (Autobahns), which decompose the graph into a collection of subgraphs and apply convolutions that are equivariant to each subgraph’s automorphism group. The specific architecture is determined by the choices of local neighborhoods and subgraphs. The proposed formalism not only generalizes the message passing GNNs but also encompasses novel architectures. Experiments are conducted on molecular graph datasets and demonstrate that Autobahns can achieve state-of-the-art results.

**Limitations And Societal Impact:**

The authors have discussed the potential negative social impact of their work.

**Main Review:**

----------Strengths----------

(1) The idea is novel and beautiful since it naturally provides a formalism of constructing permutation equivariant GNNs (proved by Theorem 1 in the paper, although I did not check the proof details) while being much more flexible than conventional GNNs. Autobahn generalizes message passing neural networks (MPNNs) and leads us to some novel architectures.

(2) The paper is well written and reasonably polished. Overall I find the paper not hard to follow.

(3) The paper provides us deeper insights on the formulation of GNNs and opens up a potential new viewpoint. As discussed in the conclusion section, I agree it is an interesting question to ask how certain choices of substructure influence the expressive power of resulting architectures, i.e., the inductive bias problem. Although it is not the scope of this paper, this work can motivate future research on the inductive bias problem of GNNs.

----------Weaknesses----------

(1) Since the construction of an Autobahn network requires solving a subgraph isomorphism problem in general, the efficiency of Autobahn might be a critical issue. However, I think more details on the efficiency of Autobahn could be provided. Moreover, I think it is interesting to know how the choice of the heuristic algorithm, which solves the subgraph isomorphism problem, affects the overall performance.

----------Overall----------

Overall, I think this paper presents a novel and interesting formulation of GNNs, which beautifully incorporate the automorphism groups of subgraphs. I like this work as it opens up a potential new viewpoint and can motivate future research. Hence, I recommend acceptance.

----------After Rebuttal----------

To conclude, because I am now more concerned about the final presentation of the theoretical results and the overall performance of Autobahn, I think I should be more cautious in recommending this paper. I would prefer to lower the score to 6 and also my confidence to 2. Please see the reply below for the details.


**Time Spent Reviewing:**

5

---

> ### Author Response · Authors · 2021-08-10
> **Response to Reviewer 3yjU**
>
> We thank the reviewer for their endorsement of this work.  The connection between substructure choice and expressive power is indeed an interesting research direction.  We hope to pursue this in future work, and we certainly hope that this paper will inspire other researchers to re-examine the inductive biases of GNNs, just as the reviewer has stated.
>
> Regarding the potential weaknesses mentioned by the reviewer, we agree that while solving a subgraph isomorphism problem might initially seem daunting, in practice we don’t see this as a substantial obstacle.  First, we observe that while checking whether two graphs contain *any* common subgraph is NP-complete, checking whether a graph contains a *specific* subgraph is polynomial in the worst case.  In practice, many “real-world” graphs (such as molecules, road networks, and code graphs) are highly structured, simplifying the process of finding subgraphs even further.  Indeed, we believe these graphs are where the ability of Autobahn neurons to incorporate additional structure will have the most effect.
>
> Finally, the reviewer has rightfully alluded to extensive prior work on efficient heuristic algorithms for finding subgraphs.  As the choice of the heuristic affects the graph preprocessing but not the runtime of the network itself, we think the best way to address this would be to remove the corresponding footnote and instead explicitly discuss the issue in the main text. In particular, we can refer interested readers to the existing body of literature on heuristics for subgraph searches.  We will happily edit our paper to do so in revision.

---

> > ### Comment · Reviewer_3yjU · 2021-09-02
> > **Response to the Authors**
> >
> > Thanks to the authors and the two reviewers for the discussions. I am sorry for the late response. I have read the discussions between the other reviewers and the authors, and I find them very helpful for a better understanding of the work.
> > #### 1. Expressivity
> > At the time I wrote the original review, I was also somehow concerned about the missing theoretical analysis. However, I thought it was fine to leave that to future work, given that the proposed architecture already showed its novelty. After reading the other review and comparing the proposed Autobahn architecture to [Bouritsas et al., arXiv] and [de Haan et al., NeurIPS2020], I am also aware of the necessity of including the theoretical analysis of the expressive power in the paper. I agree with Reviewer jtWY that the acceptance should be conditioned on the added theory is included in the camera-ready version. I also share the same worry with Reviewer NsaB that the added theories are too much for the rebuttal process. Given the limited space, I am not confident whether the added theoretical results will be appropriately presented in the final version. In this regard, I feel I should be more cautious in recommending this paper.
> > #### 2. Practical efficiency
> > I would like to thank the authors for the answer and Reviewer NsaB for the clear explanation in [the discussion thread above](https://openreview.net/forum?id=NU69dglcsS&noteId=IIp2iZ8oorV). Now, I am convinced that the complexity of the subgraph isomorphism matching problem will not be a problem for the special cases in which the application of the proposed NN is specialized. However, I would address that these also limit the range of applications of Autobahn.
> > #### 3. Generalization & Performance
> > The proposed Autobahn works reasonably well, especially on the ZINC dataset. However, as Reviewer jtYW pointed out, the performance seems to be more or less on par with GSN or other methods that use the structural feature. Moreover, when we compare the performance to the entire leaderboard of OGB-MolPCBA and OGB-MolHIV datasets, the performance is mediocre and on the same level as some very simple models like GIN + virtual node. Since it is hard to assess the newly added theoretical results and the theoretical contribution to the community, I would say the provided performance numbers are not very satisfactory.
> >
> > To conclude, because I am now more concerned about the final presentation of the theoretical results and the overall performance of Autobahn, I think I should be more cautious in recommending this paper. I would prefer to lower the score to 6 and also my confidence to 2.

---

> > > ### Author Response · Authors · 2021-09-03
> > > **Response to the Reviewer 3yjU**
> > >
> > > Our paper stands on the merits of its theoretical novelty: novelty we have clearly delineated in the paper and in the review process, and which the reviewers have agreed exists in our work. Our experimental results support this by demonstrating by showing that our ideas  can be shaped into practical GNN architectures.
> > >
> > > Frankly, we feel like we have been set up for failure. We have been asked to provide additional theoretical results. We have done so, despite our misgivings on how informative the analyses requested are for Autobahn: a topic we discuss [here](https://openreview.net/forum?id=NU69dglcsS&noteId=ZfWRfGY_b6). However, now that we have provided these results our scores are being reduced because it is doubted whether we could fit them into the paper.
> > >
> > > We do not think giving theoretical results should be a precondition of acceptance: many if not most papers advancing novel GNN architectures do not provide any. However, we are confident in our ability to integrate them clearly and succinctly into the paper should it be asked of us.

---

### Author Response · Authors · 2021-08-10
**Select Results on the Expressivity of Autobahn**

Two reviewers have asked us about the theoretical expressivity of the Autobahn architecture. Initially we had intended to analyse the performance of Autobahn in a follow-up publication.  There, the additional space would allow us to perform a more comprehensive analysis of expressivity, as well as other topics such as relative parameter efficiency.  However, we acknowledge the reviewer's interest in the topic, and are also quite open to the idea that providing some preliminary theoretical results in this paper could help motivate our work. As such, we have given some results below that can be easily proven in a relatively self-contained manner.  (We have put them in a separate comment to avoid repeating ourselves between responses)  We are happy to discuss these results further.  Moreover, we would be happy to include them, along with associated proofs, in the revised version.

1. Consider an Autobahn network whose largest template has $k$ nodes. The expressivity of this network is bounded above by the $k$’th order network as described in [1].  Moreover, there exists an Autobahn network that achieves this bound.

2. Consider a GSN network with a given set of subgraphs chosen to construct initial features.  Next, consider an analogous Autobahn network where each layer’s neurons consist of (1) the GSN’s MPNN neurons and (2) Autobahn neurons operating on the subgraphs used to featurize the GSN. The Autobahn network is at least as expressive as the GSN network.  Moreover, there exists collections of GSN subgraphs such that the Autobahn network is strictly more expressive.  (Note that we cannot say that the Autobahn network is always strictly more expressive than the GSN network because of pathological counterexamples, e.g. setting all of the GSN’s subgraphs to be the subgraph of a single node.)

3. Consider a graph of N nodes.  There exists an Autobahn network whose neurons correspond to subgraphs of size $N-1$ that is able to reconstruct the full graph even if the reconstruction conjecture does not hold.  (Note that this is in contrast to many MPNN architectures e.g.GSNs, which require the reconstruction conjecture to be true for reconstruction to be possible.)

Finally, we would like to observe that these results assess the expressivity of Autobahn.  However, what they do *not* address is the efficiency of the architecture.  What makes Autobahn viable, in our opinion, is not just its ability to perform higher-order, expressive computation, but also its ability to do it *efficiently*.  For instance, consider a hidden representation of a network corresponding to “the nodes colored red, orange, yellow, green, blue, indigo, and violet form a path.”  It is possible to store this information in a permutation-equivariant signal by considering a seventh order feature.  For instance, one could list all possible permutations of the seven nodes from an arbitrary initial ordering, and write down a $1$ for the single permutation that achieves the correct order and a $0$ for all the rest.  However, this requires a datastructure with $7!$ numbers.  In contrast, Autobahn just stores the sequence ($7$ numbers) along with a note saying “by the way, this is a path.”  This representation isn’t more expressive than the permutation-equivariant one.  However, it is much more efficient.

However, we believe that existing theoretical tools for understanding GNN are comparatively less mature than those for expressivity.  To comprehensively analyze the efficiency gain of Autobahn, we believe that we will need to develop new theoretical tools for understanding the efficiency of equivariant GNNs.  While this is beyond the scope of this paper, it is something we hope to pursue in future work.

**References**

[1] Maron, Haggai, et al. "Invariant and Equivariant Graph Networks." International Conference on Learning Representations. 2018.

---

> ### Comment · Reviewer_jtWY · 2021-08-19
> **Additional questions regarding the theoretical results**
>
> I am happy to see that the authors have derived an extensive theoretical analysis of their method. In my opinion, this was really missing from the paper. There are a few questions that arise from this new evidence and deserve a deeper examination from the reviewers:
>
>  - What are the conditions for an Autobahn network to achieve the bound in 1.? How many templates do we need to consider? Can we say anything about the k-Autobahn expressivity in comparison to the k-1 WL?
> - Theorem 2 is intuitive (this is something I expected as I mentioned in my initial review), but it is informal. Can the authors explain what do they mean by “each layer’s neurons consist of … to featurize the GSN.”? When is Autobahn strictly more expressive than GSN?
> - Does 1. and 2. imply that a GSN whose subgraphs have size of at most $k$ is at most as expressive as k-WL?
> -  Could the authors provide proof sketches? Even the high-level rationale will be fine.
> - I am not sure I follow the discussion on efficiency (mainly the last paragraph). Could the authors elaborate more on that?
>
> I understand that probably the above is too much to ask. However, my concern is that the new evidence that the authors are bringing might be significantly modifying the initial submission, and hence we as reviewers do not have all the required details to carefully assess the contributions. I believe that the above theoretical results might be necessary to recommend acceptance, hence it will be quite helpful if the authors can provide more details.

---

> > ### Author Response · Authors · 2021-08-24
> > **Response to questions regarding theoretical results**
> >
> > The reviewer’s requests are certainly not too much to ask: this is a very interesting topic and we are very happy to have the opportunity to discuss it further. Below, we have given brief answers to the reviewer’s questions, followed by sketches of the proofs.
> >
> > We also have a general comment about this line of inquiry, which we think would be of interest to the reviewer. Some of our results above attempt to answer the question “What can $k$-WL say about Autobahn?” Indeed, the $k$-WL hierarchy is one of the most popular tools for analyzing GNN expressivity.  However, we could also turn this question on its head, and ask, “What can Autobahn say about $k$-WL?” In fact, we think that Autobahn shows the limits of $k$-WL as a tool for understanding GNN architectures.
> >
> > In our opinion, the most interesting Autobahn architectures – both theoretically and practically – are the ones that are least describable using the $k$-WL hierarchy. A comprehensive exploration of this idea would certainly be too long to fit into our current paper, and we are saving a more formal treatment for later work. However, since this topic seems to be of interest to the reviewer and is relevant to our current discussion, we have attached a separate comment exploring this line of thought.
> >
> > ## Response to the Reviewer’s questions
> >
> > 1. As one might expect, a sufficient condition is for the templates used to cover all ordered tuples of k or less nodes. This can be achieved by setting the templates to all edge-less graphs of k nodes or less. This _particular_ architecture would then be as expressive as $k$-WL by the results in [2].  One could also ask “given an Autobahn template with an (unknown) template made of k nodes, can we say anything about the expressiveness of the resulting architecture with respect to $k-1$-WL?” Our answer is “No, and this is probably a good thing.” (We discuss this train of thought in the attached comment.)
> >
> > 2. With this phrase, we are saying that the set of templates used by the Autobahn network is the union of two sets. The first set is the set of all star graphs (i.e. the template formed by the action of an MPNN, as per Figure 1). The second is the set of all subgraphs that are counted by the GSN. The neurons on the star graph operate in exactly the same manner as the GSN’s MPNN layers. For the Autobahn neurons on the remaining templates, it is sufficient for their inputs and outputs to be node features.  To answer the question, “When is Autobahn strictly more expressive than GSN?”, we establish the existence of a specific case by leveraging the $k$-WL hierarchy. Under what conditions this holds in general is an interesting question. However, since both GSNs and Autobahn typically don’t fit neatly in the $k$-WL hierarchy, we believe that answering this may require the development of new theoretical techniques.
> >
> > 3. While 1 and 2 do not demonstrate this by themselves, this statement would follow as a corollary when combined with the results presented in [1].
> >
> > 4. The discussion was merely intended to point out that expressiveness is only part of the picture when assessing GNN architectures. Instead, we must consider the trade-off between expressiveness and computational requirements: two architectures may be able to learn the same function, but one might require a considerably smaller computational budget to do so. Most results on GNNs that we are aware of are binary in nature (i.e. task can / cannot be completed) but do not account for how expressiveness changes with computational cost. (Admittedly, some of our typos probably didn't help. For instance, the last paragraph should read “However, we believe that existing theoretical tools for understanding GNN _efficiency_ are comparatively less mature than those for expressivity.”)
> >
> > ## Proof Sketches
> >
> > 1. The proof is tedious, but relatively straightforward. To show the bound exists, observe that by Cayley’s theorem every Autobahn neuron can be written as a function of a subgroup of $S_k$ acting on the corresponding nodes. Moreover, $S_k$ is isomorphic to the set of all ordered tuples of these nodes, which is precisely the internal representation of the $k$’th order network. By including input features that correspond to subgraph isomorphism classes, Autobahn convolutions, narrowing, and promotion then correspond to specific operations on the terms in the $k$’th order network. To show the bound is tight, it is enough to give a single set of templates that “recovers” the $k$’th order network. This can be achieved by choosing a set of templates corresponding to all edgeless graphs of size $k$ or less, which enumerates all possible tuples of $k$ nodes or less. One then remains to show that Autobahn can perform the operations of index-permutation, tensor contraction, and tensor expansion which generate the operations inside a $k$’th order network.
> >
> > 2. Showing the Autobahn network is at least as powerful as the GSN network is simple: the GSN network is a specific Autobahn network that only uses the non-MPNN neurons in the first layer, and then only to send a unique label to each node. (Since each isomorphism-class gets its own parameters, this is easy to do.) To show that it is possible for Autobahn networks to be more powerful, we again give a specific example. We again consider templates corresponding to the edgeless graph. Each node in a network is in the same number of subgraphs isomorphic to the edgeless graph, so this does not improve GSN expressiveness over 1-WL. However, by the results from theorem 1, the resulting Autobahn is $k$-WL powerful.
> >
> > 3. An algorithm for reconstructing the graph can be explicitly constructed. (In fact, it closely resembles the procedure in Figure 3.) In the first layer, output a “1” for each edge in every neuron’s local graph. Then, perform one round of narrowing and promotion. Now, pick an arbitrary neuron in the second layer. The only nodes with nonzero incoming node features are the ones that were connected to the missing node, which reconstructs the graph.
> >
> > [1] Geerts, Floris. "The expressive power of kth-order invariant graph networks." arXiv preprint arXiv:2007.12035 (2020).
> >
> > [2] Maron, Haggai, et al. "Invariant and Equivariant Graph Networks." International Conference on Learning Representations. 2018.

---

> > > ### Author Response · Authors · 2021-08-24
> > > **Thoughts on the relationship between Autobahn and $k$-WL**
> > >
> > > The theorems we have given above establish a sort of theoretical “base camp.” They demonstrate that expressive Autobahn architectures exist and help contextualize our work in terms of previous analyses, in particular the WL hierarchy. We recognize their value as such. However, we had initially omitted them from our initial paper (perhaps mistakenly so) because we are not content with them, but would instead prefer even stronger analyses. This is because we think Autobahn is most interesting – both practically and theoretically – precisely when it is _least_ describable using the WL hierarchy. This is because Autobahn is able to selectively "borrow" the power of high-$k$ WL algorithms. We have not included this line of thought in our paper because we think it seemed a bit too heady for a general audience and could substantially add to the length. Moreover, we think the ideas inherent in it deserve a more careful treatment in their own paper. However, it is quite relevant to the topics we are discussing in the review, and we think it might be of interest to the reviewers.
> > >
> > > To understand our claim statement, let us take a step back and first consider the $k$-WL algorithm and the $k$’th order GNN architecture from [2]. In each of these approaches, we consider every possible ordered tuple of $k$ nodes. As we increase $k$, we consider larger and larger tuples, which makes the algorithm more and more powerful. However, the number of tuples, and consequently the algorithm’s cost, also increases exponentially with $k$. In practice, this means that we can feasibly run $k$’th order networks for $k=2$ or maybe even $k=3$ for small graphs.  However, they become impossibly expensive when $k$ is 4 or larger.
> > >
> > > Our attempts to increase expressiveness by simply increasing $k$ to $k+1$ have quickly hit a computational wall. Instead, we pursue a different strategy: rather than consider _all_ of the ordered tuples of size $k+1$, we can only consider a few tuples. While this won’t be _as_ powerful as the full $k+1$-WL algorithm, we have succeeded in borrowing a little bit of its power. Indeed, we probably don’t even _need_ the full power of the $k+1$-WL: we just need enough of its power to perform the learning task we care about. It is quite likely that many tuples of length $k+1$ are unnecessary or even irrelevant. At best, including them will inflate our computational requirements for no reason. At worst, they might even detract from our network’s inductive bias by bombarding it with irrelevant information.  After all, “more expressive” is not always good in Machine Learning.  Fully connected layers are much more expressive than convolutional neural networks, but they are also worse at image recognition.
> > >
> > > From this perspective, a “good” GNN isn’t necessarily a highly expressive one, or even one that achieves a specific threshold of WL expressiveness. Instead, it would be the one that intelligently selects tuples so that it can solve the learning problem in a clean, computationally efficient manner. Implicitly, this is what is attempted by most (if not all) approaches to constructing computationally-feasible higher-order GNNs. For instance, References [2,3,4] choose tuples by finding nodes and edge neighborhoods in the graph.
> > >
> > > What the Autobahn formalism contributes is a way to greatly increase the palette of ways we can choose these tuples.  Autobahn allows us to choose the nodes used in the tuples based on the isomorphism class of associated subgraphs, a much more flexible criterion than used in the aforementioned networks.  Moreover, the automorphism-equivariant convolutions give as precise control over the tuple’s internal ordering as possible without breaking global permutation equivariance. This allows us to efficiently achieve very high-order computation. Circular convolution over a ring of 8 nodes allows us to work with an 8-element tuple at exponentially less cost than that required by an 8’th-order network.
> > >
> > > In the blog post titled, “Beyond Weisfeiler-Lehman: using substructures for provably expressive graph neural networks,” one of the authors of [1] observed that GSN’s don’t quite fit into the WL hierarchy. We believe that where GSN shows the cracks in the WL hierarchy, Autobahn breaks through it. One can use Autobahn to construct GNNs that are powerful networks for a given learning task, but don’t even achieve 1-WL expressivity. For instance, consider an Autobahn whose templates are only collections of cycles. Clearly, this GNN doesn’t even achieve 1-WL: it can’t tell one tree graph apart from another. However, it can easily count large rings, a task high on the $k$-WL hierarchy. And this is not just a “toy” example.  When applied to certain families of graphs such as irregular lattices, it would likely achieve very strong performance in practice.
> > >
> > > Returning to the question of theoretical analysis of GNNs, we think that some of the most interesting neural architectures may be ones with expressiveness _in between_ the rungs of the WL hierarchy. Understanding architectures in this liminal space is, in our opinion, an interesting open question in GNN research. For instance, one approach could be to construct WL-like algorithms by considering different families of node tuples, which could lead to new, complementary classification systems for GNNs. Moreover, the analogy with Fully-connected layers and Convolutional layers suggests that merely analyzing expressivity may not be enough by itself.  We also need more research in ways of analyzing how well or poorly suited GNN architectures are for particular graph families.  Ultimately, this is a stimulating research direction. We hope to pursue this in future work, and we hope the ideas inherent in our current paper will inspire others to do the same.
> > >
> > > [1] Bouritsas, Giorgos, et al. "Improving graph neural network expressivity via subgraph isomorphism counting." arXiv preprint arXiv:2006.09252 (2020).
> > >
> > > [2] de Haan, Pim, et al. . “Natural Graph Networks” Advances in neural information processing systems (2020).
> > >
> > > [3] Kondor, Risi, et al. "Covariant compositional networks for learning graphs." arXiv preprint arXiv:1801.02144 (2018).
> > >
> > > [4] Maron, Haggai, et al. "Invariant and Equivariant Graph Networks." International Conference on Learning Representations. 2018.
> > >
> > > [5] Vignac, Clement, et al. "Building powerful and equivariant graph neural networks with structural message-passing." arXiv preprint arXiv:2006.15107 (2020).

---

### Decision · Program_Chairs · 2021-09-27

**Decision:**

Accept (Poster)

**Comment:**

The paper proposes a novel structure-aware GNN architecture based on partitioning the graph into subgraphs.
The authors provided extensive responses to the reviewers' comments in the rebuttal and followup discussion. There was also an extensive discussion among the reviewers. The reviewers recognize the novelty of the paper. While after rebuttal and discussion some reviewers believe the paper still requires significant editing to improve the clarity, the AC is of the opinion that the novelty outweighs these drawbacks and would like to give the authors the benefit of the doubt recommending to accept the paper and hoping the authors will improve the presentation and introduce edits according to the reviewers' suggestions.